# Hippocampal adult-born granule cells drive network activity in a mouse model of chronic temporal lobe epilepsy

F. T. Sparks [1,2,3,5], Z. Liao[1,2,3,5], W. Li[1,2,3], A. Grosmark[1,2,3], I. Soltesz [4] & A. Losonczy [1,2,3✉]

Temporal lobe epilepsy (TLE) is characterized by recurrent seizures driven by synchronous neuronal activity. The reorganization of the dentate gyrus (DG) in TLE may create pathological conduction pathways for synchronous discharges in the temporal lobe, though critical microcircuit-level detail is missing from this pathophysiological intuition. In particular, the relative contribution of adult-born (abGC) and mature (mGC) granule cells to epileptiform network events remains unknown. We assess dynamics of abGCs and mGCs during interictal epileptiform discharges (IEDs) in mice with TLE as well as sharp-wave ripples (SPW-Rs) in healthy mice, and find that abGCs and mGCs are desynchronized and differentially recruited by IEDs compared to SPW-Rs. We introduce a neural topic model to explain these observations, and find that epileptic DG networks organize into disjoint, cell-type specific pathological ensembles in which abGCs play an outsized role. Our results characterize identified GC subpopulation dynamics in TLE, and reveal a specific contribution of abGCs to IEDs.

[1] Department of Neuroscience, Columbia University, New York, NY, USA. [2] Mortimer B. Zuckerman Mind Brain Behavior Institute, Columbia University, New York, NY, USA. [3] The Kavli Institute for Brain Science, Columbia University, New York, NY, USA. [4] Department of Neurosurgery, Stanford University, Stanford, CA, USA. [5]These authors contributed equally: F. T. Sparks, Z. Liao. ✉email: al2856@columbia.edu

Temporal lobe epilepsy (TLE) is a common neurological disorder characterized by recurrent focal seizures originating in the mesial temporal lobe, most commonly the hippocampus (HPC). While the circuit mechanisms of TLE are not well-understood, it is thought that the pathological features of TLE emerge from a complex series of microlevel reorganizational steps leading to persistent perturbation of the excitatory and inhibitory processes regulating entorhinal–HPC interactions[1–4]. Hypersynchronous macrolevel network events associated with TLE, in turn, are thought to be primarily the result of such microlevel alterations to underlying network connectivity within the HPC formation[5–9].

Macrolevel synchronous events in epilepsy can be identified in electroencephalography (EEG) or local field potential (LFP) recordings as large amplitude and long duration electrographic seizures[10,11], high frequency events[12], and most commonly and frequently, interictal epileptiform discharges (IEDs)[13–15]. These EEG events are thought to arise from pathophysiological changes to excitatory and inhibitory microcircuits in the epileptic network[16]. IEDs are transient EEG events characterized by a short duration (<100 ms) and large amplitude, and include multiphasic discharges as well as single interictal spikes[17]. At the macrolevel, these events appear to be the result of widespread, recurrent, synchronous population firing, though recent investigations into population dynamics at the circuit level have revealed heterogeneity of individual neuron responses during IEDs[2,18]. This suggests that relatively sparse sub-ensemble dynamics support IEDs, and that these dynamics may contribute to the diversity of epileptiform activity identified in EEG or LFP recordings. These observations have been corroborated in human epilepsy patients via multiunit activity recordings from depth electrodes[19]. In addition, observations supporting this hypothesis have been made in animal models of TLE by imaging excitatory granule cells (GCs) of the dentate gyrus (DG) in an in vitro slice preparation[2], and imaging either excitatory or inhibitory cell types in CA1 in vivo[1]. While both excitatory and inhibitory cell classes have been shown to be involved in IEDs, how the collective activity of heterogeneous principal cell types gives rise to local network dynamics during pathological activity in chronic epilepsy in vivo, especially in the upstream nodes whose outputs shape CA1 excitation–inhibition dynamics, remains incompletely understood.

As the entry node of the HPC, the DG plays a critical role in cognitive processing by regulating the propagation of HPC feedforward drive[20]. At the microcircuit level, strong local inhibition and lack of recurrent excitation of GCs result in the sparse GC population activity that is thought to support computational functions such as pattern separation and novelty detection[21,22]. This intrinsic low excitability is also thought to enable the DG to restrict the relay of synchronous activity from the entorhinal cortex into the HPC, thereby regulating the propagation of excitatory activity, a property known as dentate gating[23,24]. The breakdown of the DG gate is hypothesized to contribute to epileptogenesis, leading to seizure generalization as well as cognitive and memory deficits[25,26]. In the adult DG, new GCs are continually generated and functionally integrated into DG circuitry[27–29]. This adult-born GC (abGC) subpopulation of the DG network is especially sensitive to seizure-induced reorganization[30–35], that may take the form of an altered level of neurogenesis[36–38], mossy fiber sprouting[33,39–43], abnormal formation and persistence of basal dendrites on abGCs[31,32,44,45] as well as ectopic dispersion and migration of abGCs into the hilus or CA3[30,46,47]. A recent in vivo study of normal DG function demonstrated that abGCs are intrinsically more active and less stimulus-selective than their mature counterparts (mGCs)[48]. Together, these findings strongly implicate abGCs in the development of epilepsy, and support a hypothesis in which the relative hyperactivity of abGCs is amplified through recurrent microcircuits in the epileptic DG, resulting in generalized synchronous seizure activity in TLE[31,34,35,49–51]. However, at present there are no in vivo data allowing direct comparisons of the activity patterns of major GC subpopulations and their relative contribution to functional network structure during macrolevel epileptic events.

In the present study, we genetically label populations of mGCs and abGCs in the HPC, induce the intra-HPC kainic acid (KA) model of chronic TLE, and then dissect the activity dynamics of these neural populations using a combination of in vivo two-photon calcium imaging and LFP recording. To uncover ensemble structure underlying high-density calcium recordings of the chronically epileptic DG network, we introduce a generative model framework for ensemble recruitment that captures existing knowledge and biological intuitions about the mechanisms of IEDs. Performing statistical inference on this class of model is difficult in general, but we show that our biologically motivated model can be reduced to Latent Dirichlet Allocation (LDA), a well-known topic model with many tractable inference algorithms[52]. We use this reduction, in combination with tools from the LDA literature, to infer the recruitment of microensembles by network-level events. Consistent with previous in vitro functional imaging from the DG of slices taken from chronically epileptic animals[2], we found structured ensemble dynamics within the GC population during IEDs in vivo. In addition, we also identified cell-type specific ensemble structure nested within the abGC and mGC populations, with distinct contributions to network dynamics in the interictal period. Finally, we found that abGC ensembles participate in network activity during IEDs to a disproportionately higher degree than mGCs, and in a desynchronized manner compared to activity during physiological high frequency electrographic events i.e., sharp-wave ripples (SPW-Rs)[53]. These observations suggest that the chronically epileptic DG network exhibits robust underlying functional organization in temporal correlation and GC lineage. Because abGCs are strongly reorganized in TLE compared to their mature counterparts[30–33], our results suggest that this reorganization produces functionally distinct ensembles that have different pathophysiological roles identifiable in vivo.

## Results

**Two-photon calcium imaging of the epileptic DG in vivo.** Dissecting the functional and anatomical microcircuit structure of the DG in vivo requires techniques that allow simultaneous recording of neural activity from a wide section of the network at single-cell resolution, while also permitting molecular identification of the recorded cells. This was achieved by performing two-photon calcium imaging of the DG in head-fixed mice on a circular treadmill[48]. Combining these techniques with a cell-type specific *Cre* driver line allowed us to indelibly label and image abGCs and mGCs simultaneously in a single recording[48] (Fig. 1a, b), which is not feasible with electrophysiological recordings or in freely moving animals. To indelibly label abGCs, NestinCreER[T2] mice were crossed with a conditional reporter line (Ai9) and pulsed with tamoxifen (TMX) to express the fluorescent red reporter (tdTomato) in young abGCs (abGCs were 5 weeks old at the start of imaging, Fig. 1c, see "Methods" section). Mice were then stereotactically injected in the dorsal DG with a recombinant adeno-associated virus (rAAV) to express GCaMP6f in all GCs. KA was injected unilaterally into the ventral HPC ipsilateral to the viral injection site to induce *status epilepticus*, simulating the initial insult that leads to epileptogenesis in humans[54,55]. The KA injection site was chosen to minimize associated gliosis and cell loss within the imaging field of view,

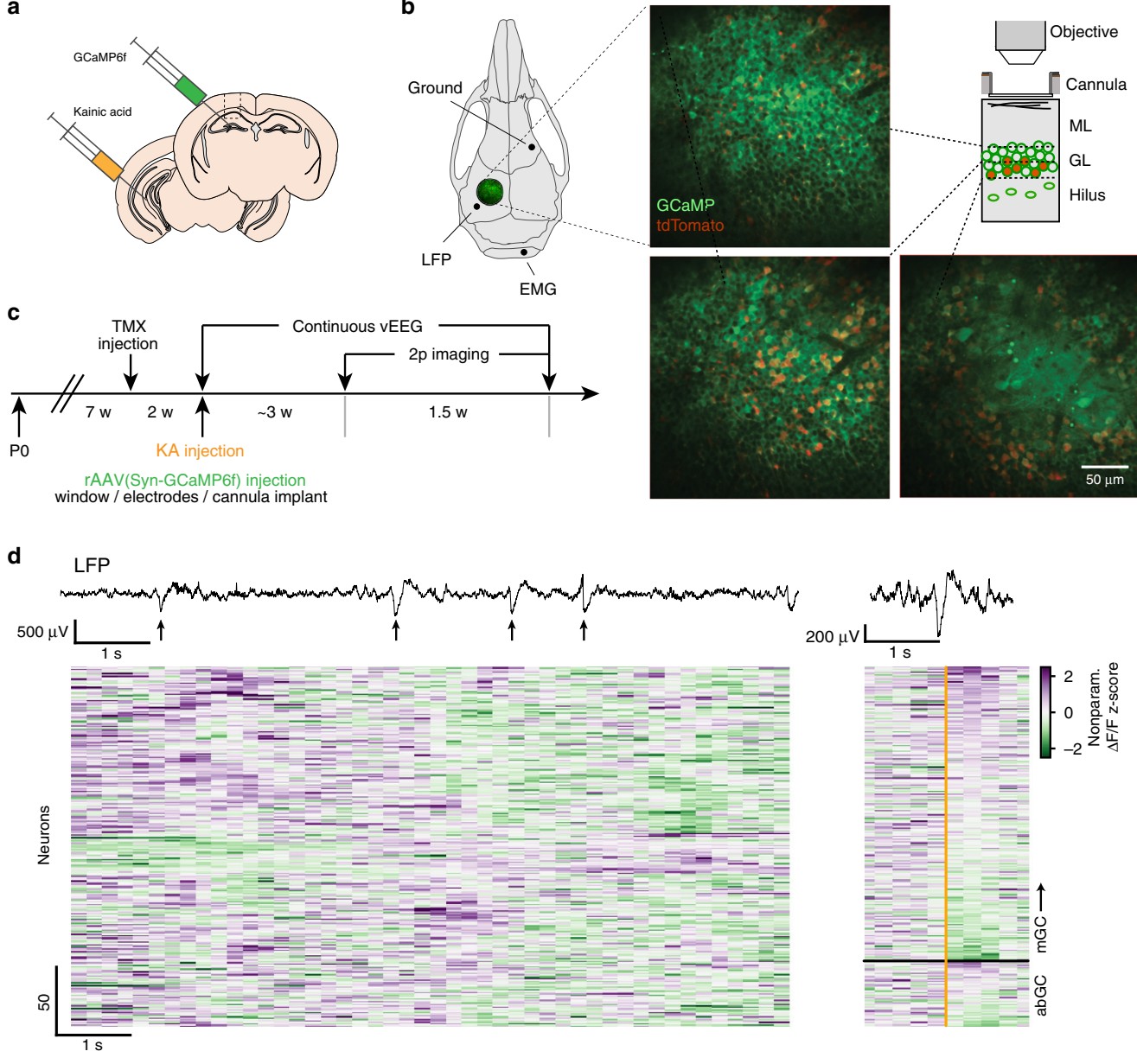

**Fig. 1 Cell-type specific functional imaging in the dentate gyrus. a** Experimental schematic. GCaMP6f is virally expressed in all DG neurons at the dorsal HPC site of injection, and kainic acid (KA) is injected into the ipsilateral ventral HPC to induce the model of epilepsy. **b** Left: Schematic of skull showing locations of imaging window, ipsilateral local field potential (LFP) electrode, electromyography (EMG), and ground electrode placements. Center: Two-photon line-scanning microscopy allows for the recording of large populations of GCs in surgically exposed dorsal DG. Representative example in vivo two-photon two-color microscopy time average images from near simultaneous multi-plane imaging in the mouse DG. The top two planes are focused in the dorsal GC layer while the bottom plane is located just above the polymorphic hilar region. Green represents GCaMP fluorescence in all cell types, while red represents tdTomato expression limited to abGC in a Cre-dependent manner. Right: Schematic showing near-simultaneous multiplane imaging throughout the DG granule cell layer. ML molecular layer, GL granule cell layer. **c** Experimental timeline for the labeling of abGCs, induction of the kainate model, and two-photon imaging of the epileptic DG network (n = 5 mice). Adult-born GCs were indelibly labeled with tdTomato following injections of tamoxifen (TMX) that drove expression of Cre in Nestin+ cells. Two weeks later, KA was injected into the ventral HPC and GCaMP6f was injected into the dorsal DG and chronic imaging window cannula was implanted above the dorsal DG. Following injection of KA, mice were monitored for ictal activity using continuous video-EEG. **d** Top: Filtered LFP showing interictal epileptiform discharges (arrows) in an IHK mouse 31 days after KA injection. Left: Example abGC and mGC whole population activity during IEDs across a 10 second time window. Activity for each neuron is shown as a positive (purple) or negative (green) nonparametric z-scored ΔF/F, clipped to ±2.5z for visualization. Right: LFP trace and population activity from the same abGCs and mGCs centered on an example IED (orange) across a 2 s time window and sorted by maximum activity.

thereby permitting imaging of the DG within the vicinity of the main insult to the network[55]. Three days later, a chronic imaging window was implanted over the dorsal DG providing optical access necessary for cell-type-specific imaging of the GC layer in the dorsal blade of the DG. An ipsilateral single-channel LFP electrode was also implanted in the HPC ipsilateral to KA injection targeting the DG, and an electromyographic electrode implanted in the cervical trapezius neck muscle (Fig. 1b, c). After recovering from the implant surgery, mice were continuously video-EEG monitored, and TLE onset was determined by the

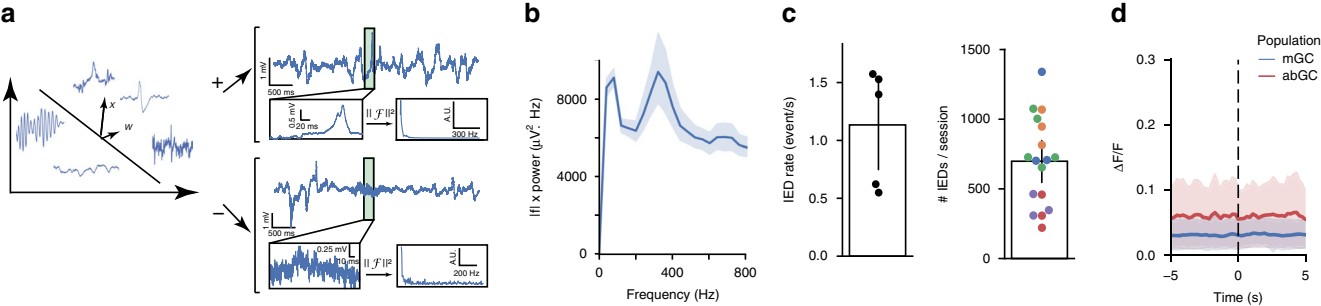

**Fig. 2 Interictal spike identification and basic characterization. a** Classifier schematic. Left: An Online Kernelized Perceptron was trained on the LFP magnitude spectrum where the window size was locked to each imaging frame. Right: Examples of classified interictal spike (above) and non-spike (below) highlighted in green, with expanded view (below left) and magnitude spectrum (below right). **b** Average IED magnitude spectrum. Average magnitude spectrum of $n = 459$ events from one subject, with $1/f$ power correction applied (data presented as mean value $+/-$ 95% CI). A peak frequency occurs in the 250–400 Hz range, exaggerated by the $1/f$ correction, reflecting the timescale of positive and negative deflections (on the order of 10 ms in an event lasting ~50 ms), which may be polyphasic for some IEDs. **c** Summary of detected IEDs rate across mice ($n = 5$ mice). Left: Average IED rate, mean 1.13 IED/s, standard deviation 0.53 IED/s. Each point corresponds to a mouse (average of 3–5 sessions per mouse). Right: Total IED count per 10-min recording session, mean count = 680 IEDs/session, standard deviation = 319 IEDs/session (each point corresponds to a single session, each color corresponds to a different mouse). Error bars: $+/-$ 95% CI in the mean, calculated from 1000× bootstrap sampling on recording sessions with replacement. **d** Spike-triggered PSTHs showing averaged response to IED by cell type, pooled across five animals ($n = 239$ abGCs, red; 1736 mGCs blue). The majority of GCs are not modulated by individual spikes on average (data presented as mean value $+/-$ 95% CI).

occurrence of the first detected generalized motor seizure event, which occurred on average 10 days after KA injection (242.9 ± 20.4 h, mean ± s.d.) in $n = 5$ mice.

Given that the laminar organization of GCs in the dorsal DG is parallel to the optical field of view[56], we can capture a large number of GCs within the GC layer and the sub-granular zone (Fig. 1b). To further increase simultaneous imaging of multiple cell-type specific populations across these layers, we coupled our image acquisition control to a piezoelectric crystal for fast axial focusing and near simultaneous multiplane imaging[57] (Fig. 1b). With multiplane imaging, we were able to simultaneously capture on the order of 433 ± 261 mGC and 56 ± 31 abGC (mean ± s.d.) active regions of interest within a single subject (example LFP recording, population $\Delta F/F$ to series of IEDs and example traces triggered by one IED shown in Fig. 1d, note that only a fraction of the GCs participate in any given IED; see below). In total, across five mice, 2164 mGCs (tdTomato-negative) and 278 abGCs (tdTomato-positive) (Table S1) were identified as active GC regions of interest during three consecutive 10 min long imaging sessions for each mouse (see "Methods" section).

**Time domain, frequency domain, and mean calcium response of IEDs**. The low probability of capturing a spontaneous seizure event during a two-photon imaging session poses a challenge to the optical dissection of epileptic networks in vivo. However, IEDs provide a series of functional "snapshots" of discrete components underlying the ictogenic circuits, during which only a subset of the network is activated[1,2]. Therefore, we sought to analyze network activations during IEDs with the aim of first understanding these "snapshots" individually, and ultimately assembling them into a complete tapestry of the network. To this end, we segmented the LFP into time bins matching the imaging frames, and classified the imaging frames during which IEDs occurred. Traditionally, IEDs are classified by hand, based on morphological properties such as event width, deflection amplitude, and deflection shape; however, annotating these features for each event manually is labor-intensive. We developed a procedure for semi-supervised IED classification and detection to generalize these principles (Fig. 2a). Because all information about the width, amplitude, and morphology of each event is contained in its frequency-domain representation, we trained a classifier on the power spectrum of a small set of hand-identified IEDs and allowed it to generalize and

update its predictions in an interactive manner; final classification results were verified for correctness by the experimenters (see "Methods" section). Other types of interictal events, such as high-frequency oscillations were not included by the classifier and were thus excluded from further analysis (Fig. 2a). Consistent with previous work[58], we found morphological heterogeneity within the classified IEDs. Despite this heterogeneity, frames containing IEDs exhibit a stereotyped LFP power spectrum after applying a $1/f$ correction, supporting the reliability of this classification method (Fig. 2b). We find an average IED rate of 1.13 ± 0.53 IED/s, with an average of 680 ± 319 IEDs per 10-min recording session (mean ± s.d.) (Fig. 2c). We then sought to quantify the modulation of single cells by IEDs. It has been hypothesized that IEDs are primarily driven by abGC activity[59], and indeed we found that abGCs are significantly more responsive to IEDs on average compared to mGCs ($p < 0.05$, Wilcoxon signed-rank test), though both are close to 0 (Fig. S1A), without significant differences in the responses of the two populations to onset ($p = 0.89$) or offset ($p = 0.34$) of locomotion (Fig. S1A). Nevertheless, we found that the mean IED-triggered PSTH was flat for both abGCs and mGCs (Fig. 2d). This observation is important as it establishes that at least one of two conditions must hold: either (a) most IEDs only recruit a sparse subset of cells (in which case the "average" cell's response to any particular IED would be small), or (b) most cells only respond to a sparse subset of IEDs (in which case any particular cell's response to the "average" IED would be small). In the remainder of the paper, we show that not only are both (a) and (b) true, but we can also identify subsets of cells that are consistently recruited together, thereby relating macrolevel LFP activity to the activity of individual neurons in a local network.

**Different IEDs recruit different populations of GCs**. IEDs represent heterogeneous, global events in the epileptic brain[58,60]. Given that we can only observe IEDs in a local network, treating them in aggregate as if they are homogeneous obscures the microlevel population activity patterns underlying each individual IED. We used logistic regression to ask the question of whether abGCs vs mGCs exhibited different IED response profiles: by training a logistic regression linear classifier to decode cell identity based solely on the cells' IED responses, we assessed whether calcium responses of abGCs and mGCs differed on an IED-by-IED basis (Fig. 3a). This classification procedure is equivalent to

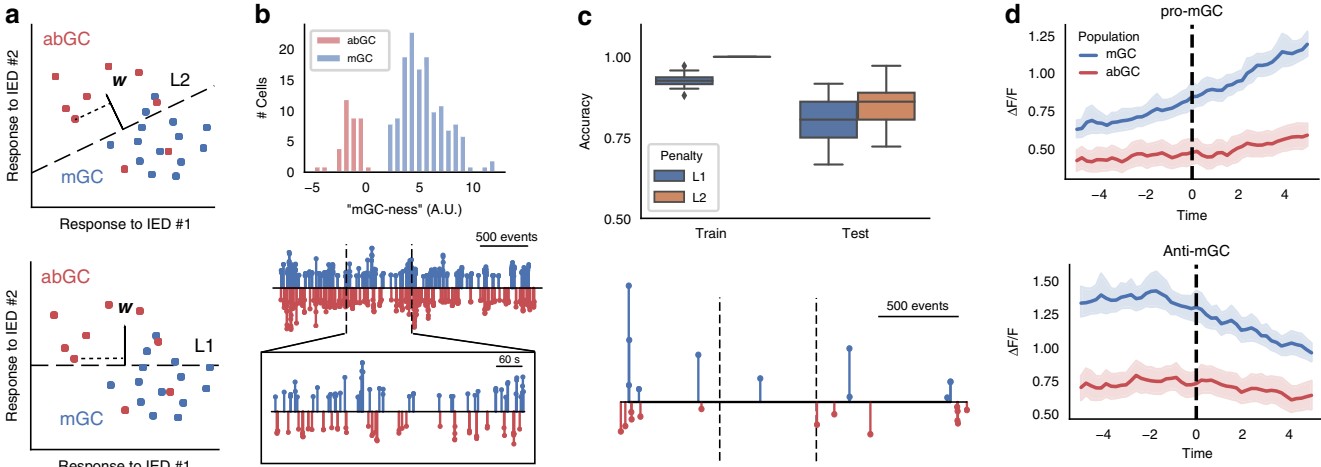

**Fig. 3 Linear classifier shows differential participation of abGC and mGC populations during interictal spikes.** Individual IEDs preferentially recruit abGC (red) or mGC (blue) neurons ("preferred" = positively modulated). **a** Schematic of L1 and L2 embedding/classification procedure. Logistic regression projects the "response fingerprint" of every cell down to a maximally separating line in spike-response space (*w*). L1 regularization introduces sparsity to *w*, selecting the fewest events necessary to perform accurate classification. **b** Top: The 1D projection as an "mGC-ness" score (top histogram). Middle: The embedding weights (stem plot, A.U.), plotted by event number over a 30 min recording session. Expansion: IEDs in the middle 10 min of the session, plotted across time. **c** Logistic regression with L1 ("lasso"; blue) regularization reveals a sparse subset of interictal spikes that is "most informative" about cell identity. Top: 100 times cross-validation on an 80–20 train-test split shows 85 ± 5% classification accuracy on the test set under standard L2 (orange) regularization and 81 ± 5% classification accuracy under L1 regularization, implying that a small subsets of IEDs are associated with either mGC or abGC activation, such that cell activity within these events alone is sufficient to accurately predict cell identity. Center line, median; box limits, upper and lower quartiles; whiskers, 1.5× interquartile range. Bottom: Similar to **b**, weights associated with "most informative" events (A.U.) identified from lasso classifier plotted vs. event number over three back-to-back recording sessions (vertical dashed lines). Events that are predictive of mGC identity, termed "pro-mGC" events, are colored in blue, while events where mGCs tend to exhibit negative responsiveness or abGCs tend to exhibit positive responsiveness are termed "anti-mGC" events, and are colored in red. See Fig. S2 for comparison across animals. **d** PSTHs triggered on pro-mGC (left) and anti-mGC (right) IEDs for an example mouse. Pro-mGC IEDs positively modulate mGCs, while anti-mGC events negatively modulate mGCs (right, in blue). In contrast, abGCs do not appear to be significantly modulated in either direction by either event type (data presented as mean value +/− 95% CI).

projecting the original *M*-dimensional responsiveness vector for each cell (where *M* is the number of electrographically identified IEDs) down to a 1D line representing a scalar "mGC-ness" score. The histogram of scores is bimodal, i.e., mGCs (blue) are mapped to positive scores and abGCs (red) are mapped to negative scores (Fig. 3b). To perform the projection, a real-valued "weight" (A.U.) shared by all cells is calculated for each IED, that can be interpreted as the degree to which positive modulation by that IED is associated with mGC identity (positive weights, which we call a "pro-mGC" IED) *versus* abGCs (negative weights, "anti-mGC" IED). Under logistic regression, the two classes are assumed to be contrastive; that is a "pro-mGC" IED is associated with both positive modulation of mGCs or negative modulation of abGCs, and likewise an "anti-mGC" IED is associated with negative modulation of mGCs or positive modulation of abGCs. Sorting the weights (thresholded to a cutoff of ±0.1) by IED time (Fig. 3b, middle) shows that most events are associated with some pro-mGC or anti-mGC bias, though this analysis does not permit us to say whether the activations are structured, as the sequence of mGC- and abGC-predictive events appears random when treating the two populations as internally homogeneous (Fig. 3b, middle and bottom). Thus, we conclude that each IED carries a small amount of information about cell identity, but the recruitment profile of each event is stochastic as any particular cell may be positively modulated, negatively modulated, or unresponsive to any particular IED regardless of the cell's identity. However, the complete profile of a cell's response to all IEDs is sufficient to predict cell identity (i.e., abGC or mGC) with high accuracy (Fig. 3b).

We used L1 and L2 regularization of the classifier in order to further probe how cell identity influences participation in IEDs. The regularized classifier could find that participation in a particular sequence of mixed events is predictive of abGC identity, which would indicate that abGCs and mGCs are internally coherent and respond to IEDs through population-specific patterns. A second possibility is that the regularized classifier may find individual "pure" or near-pure mGC-associated IEDs, which would indicate that some IEDs are attributable almost exclusively to one population. Finally, a third possibility is that the regularization penalty results in a much worse test accuracy, which would indicate that the entire profile of a cell's response to IEDs is necessary to classify the cells accurately.

Based on the cells' IED responses, an L2-regularized logistic regression classifier achieved 85 ± 5% accuracy on a test set, balanced to remove the effect of population size, cross-validated over 100 random 80–20 train-test splits (Fig. 3c, left; Fig. S1E). Subsequently, we use a sparsifying (L1) regularization penalty in order to concurrently identify the subset of IEDs that are "most informative" about cell identity, in the sense that knowing a cell's response to those events is almost as predictive of identity as knowing the cell's complete response profile (Fig. 3c right; see Methods). The L1 regularization penalty resulted in a slight reduction in test set accuracy (81 ± 5% accuracy using the same procedure), but identified IEDs with population-specific recruitment profiles and disregarded the majority that exhibited mixed activation, suggesting the second hypothesis above holds true, i.e., cell identity can be decoded from participation in events that other cells of the same type are known to participate in (Fig. 3c; see "Methods" section). We verified the efficacy of this classification procedure with a two-way analysis of variance, that yielded a main effect for IED class (pro-mGC vs. anti-mGC; $F(3,54) = 14.11$, $p = 6 \times 10^{-7}$), such that pro-mGC IEDs significantly positively modulated mGCs compared to anti-mGC

IEDs. The main effect of population was non-significant ($F(3,54) = 0.13$, $p = 0.71$), suggesting that our observations cannot be explained by intrinsic differences in responsiveness between the two populations independent of the type of IED. However, the interaction effect was significant ($F(3,54) = 10.29$, $p = 0.002$), indicating a crossing over effect, i.e., that pro-mGC IEDs significantly positively modulated mGCs over abGCs. This verifies that single cell responses to the "most informative" IEDs identified by this procedure showed high within-population heterogeneity but striking between-population differences (Fig. S1B): in contrast to the flat response to undifferentiated IEDs (Fig. 2d), we found subsets of IEDs that significantly modulate the mGC population (both positively and negatively), while the abGC population is not significantly modulated in either direction by either event type (Fig. 3d and Fig. S1C). Thus, we conclude that most IEDs differentially recruit abGCs compared to mGCs, but the recruitment of any particular cell by any IED appears random. Reliance on only one population is allowable under the assumptions of logistic regression, as an algorithm that classifies from the mGC population with high sensitivity and specificity can achieve high accuracy overall by simply classifying "non-mGC" examples as abGCs. This limitation highlights the need for a model that can more expressively capture the activation of different populations, and perhaps functionally-defined subpopulations, by different subsets of IEDs.

**A generative model of latent ensemble recruitment uncovers within-population ensemble dynamics**. The linear decoding analyses above imply heterogeneous, cell type-related dynamics among IEDs. However, we have so far assumed that abGCs and mGCs form internally homogeneous populations, while there is a growing body of evidence that this is not the case[61]. A limitation of logistic regression in this setting is that a mix of abGCs and mGCs respond to most IEDs, but the mix itself may be structured. Even among the "most informative" pro-mGC and anti-mGC IEDs identified by the sparse decoder (Fig. 3c), there is clear heterogeneity of the responses between individual cells within a population, even within a single mouse (Fig. S1B). Furthermore, the linear decoding analysis classified abGCs as those cells that did not significantly respond to pro-mGC or anti-mGC IEDs, whereas we might also like to identify IEDs that independently modulate the abGC population. We sought to construct a generative model that relates the population activity in the imaged local network to macrolevel IEDs via the hidden functional ensemble structure within the network, that can be compared post hoc with ground truth GC identities. This model should be able to account for the ensemble structure and activation underlying the data with the following properties: (1) Identifies functional ensembles that can be compared with cellular identities; (2) Infers underlying ensemble structure incrementally revealed by successive IEDs, even if an entire ensemble is never observed to be active all at once; (3) Sliding scale of activation: ensembles may be activated to differing degrees by different IEDs; (4) Mixed membership: cells may be associated with more than one ensemble, and may be associated to differing degrees to ensembles; and (5) Parameter inference (ensemble composition, activation in each IED) from data is computationally tractable.

We developed a generative model motivated by Bayesian topic modeling[52,62], that we call Latent Ensemble Recruitment (LER), and that has all of these properties. This LER model (Fig. 4a and Fig. S2A) assumes that the network consists of $K$ fixed, unobserved ensembles of cells. Each IED recruits a sparse subset of these ensembles, and each ensemble may activate a few or many of its cells, depending on the degree to which it was excited by that IED (see "Methods" section for full description of the

generative process). Various strategies are possible for performing inference on this model, though the number of latent variables we must learn compared to the number of observables presents a challenge. One convenient inference strategy is to perform ad hoc inference on $z$ using bootstrapping to binarize the dataset ($z'$), in order to solve the resulting inference using a standard variational Bayes solver[63]. This ad hoc procedure is equivalent to reducing our model to Latent Dirichlet Allocation, a closely related model for which the inference problem has been well-studied, using the topic modeling analogies: ensemble ~ "topic", spike ~ "document", cell ~ "word"[52] (Fig. S2A). The binarized data $z'$ (Fig. 4b) contains some qualitative ensemble structure and temporally distinct activations between abGCs and mGCs. The ensemble activation matrix $\theta$ (Fig. 4c) shows temporally coherent activations (i.e., an ensemble activated by event $i$ is likely to also be activated by events $i - 1$ and $i + 1$), despite the exchangeability of IEDs (i.e., the model receives no information about the order or relative timing of events). This temporal coherence may be due to temporary increases in excitability localized to the micro-circuits underlying each IED, such that subsequent IEDs are more likely to recruit the previously excited ensembles. Figure 4d shows examples of learned ensembles $\sigma$: we say a cell "participates" in an ensemble $k$ if the weight of that cell in $\sigma_k$ is $>3/N$ (i.e., greater than $3\times$ uniform prior level). Even though the model receives no information about cell lineage, from these inferred results we can clearly distinguish mGC-predominant, abGC-predominant, and mixed ensembles (Fig. 4d). Consistent with results previously reported in vitro[2], we also find low overlap ($\leq 3$ cells) between any pair of ensembles, as measured by the Pearson correlation between ensemble vectors (Fig. S2B, top). Because of this low overlap, we can interpret the LER results as a sort on the cell-cell correlation matrix. Sorting the correlation matrix accordingly, we find that the cells within an ensemble are highly correlated to each other and less correlated to cells from other ensembles, as expected (Fig. 4e and Fig. S2B, bottom), providing independent confirmation that the ensembles discovered by the model are reasonable. Finally, we sought to determine whether the most active ensemble in each IED could be decoded from the power spectrum of the IED itself. For small ensemble number ($K$), we can decode the most active ensemble in a held-out test set significantly better than chance using a random forest classifier with 100 estimators trained on the LFP spectrum collected within an imaging frame (Fig. 4f). We conclude that IEDs are driven by multiple ensembles with temporal and lineage-dependent structure, and conversely, different ensembles are associated with distinct electrographic signatures on LFP.

**Statistics of ensemble structure across animals**. Finally, we sought to quantify the extent to which the underlying ensemble structure is homologous across animals. We fit LER to data ($n = 5$ mice) over a range of ensemble numbers $K$, with ten realizations of LER fitted per $K$, per mouse. We find that the latent ensembles are disjoint for all animals: for choices of $K$ larger than 3, the cell vectors of distinct ensembles ($\sigma$) had Pearson correlations near 0 (Fig. S2A, top). In contrast, we find relatively higher activity correlations between ensembles, as measured by Kendall's tau (Fig. S2A, bottom). For $K$ values larger than 3, off-diagonal entries in the cell correlation matrix are consistently weakly to moderately correlated; this correlation increases monotonically and reached $\tau = 0.5$ when $K = 15$. This observation validates our choice of $K = 6$ across mice, where activity correlation between ensembles is weakly positive, suggesting some contribution of multiple ensembles to each spike; selecting a larger value for $K$ than the true number of underlying ensembles would artificially split the true ensembles, resulting in spurious ensembles with

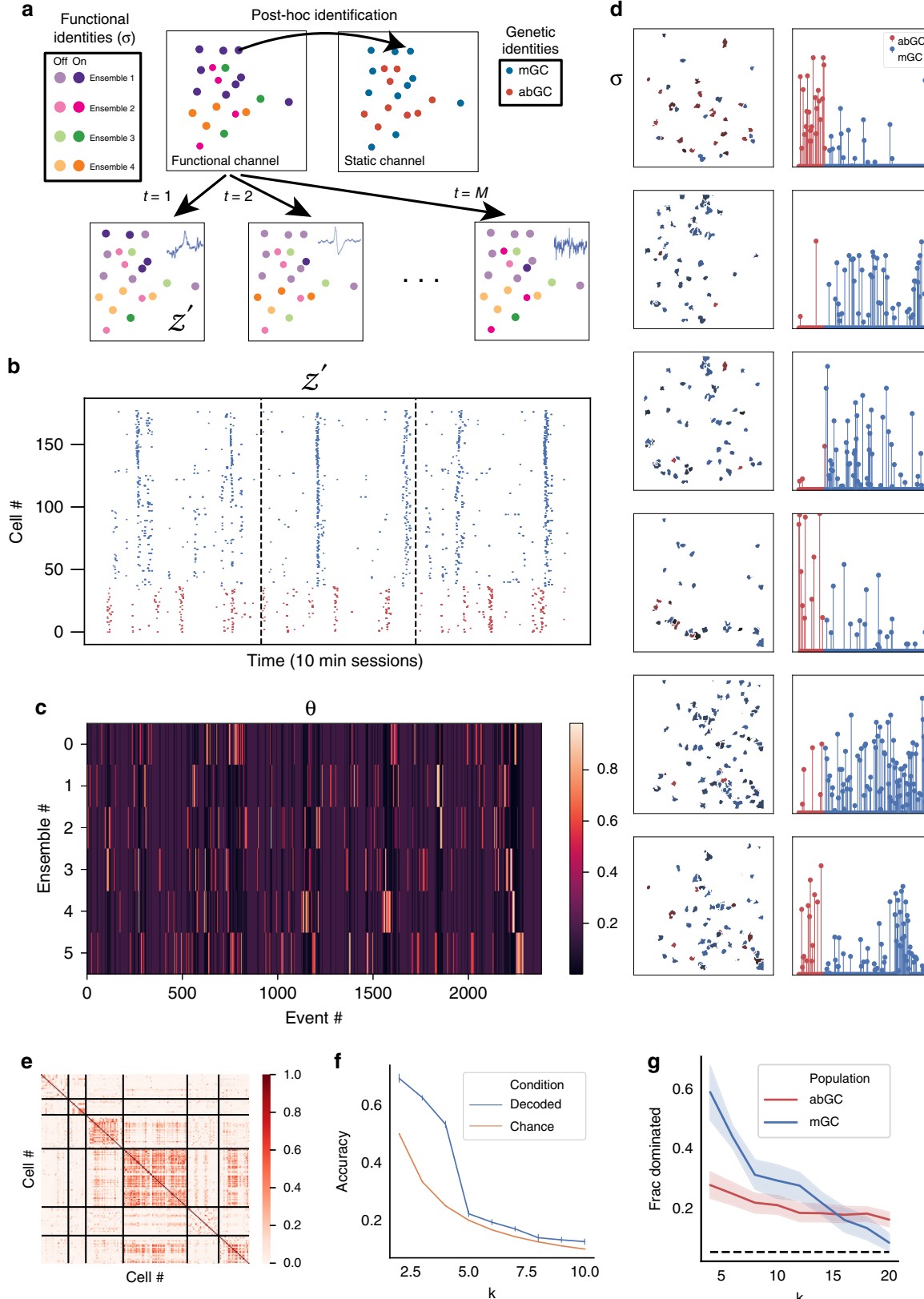

correlated activity. Importantly, we found that irrespective of the number of ensembles specified, the fraction of abGC-dominated ensembles remains fixed at 25% across mice, more than double the proportion of abGCs in the extracted population (~11%) (Fig. 4g). As abGCs themselves represent about 11% of the identified cells in our dataset, and less than 5% of GCs in the DG

network[27–29], this result strongly suggests that abGC-dominated ensembles disproportionately drive network activity in the epileptic DG during IEDs. We also compared the "purity" of the "purest" abGC ensemble to the "purest" mGC ensemble and found that this "maximum purity" (expressed as a fraction from 0 to 1) was increasing with $K$ for both populations, but nearly pure

**Fig. 4 A generative model of spike ensemble recruitment.** Latent Ensemble Recruitment (LER) model uncovers latent ensemble structure in the data. Shaded areas and error bars correspond to 95% confidence interval of the mean, calculated from 1000× bootstrap sampling on model realizations with replacement unless otherwise noted. **a** Schematic of LER generative model. Colors represent latent (hidden) ensembles in the data. Cells (colored circles) are colored by the ensemble they are most associated with. Cells in the same ensemble tend to be recruited together by IEDs (bottom). The ensemble identities can be compared to known genetic identities post hoc. **b** Raster plot of bootstrapped "activation" variable $z'$, 10,000 bootstraps with $\alpha = 0.05$, sorted by identity (red: abGCs, blue: mGCs) in synchronous network events excluding IEDs in three consecutive 10 min recording sessions, plotted against real time ($n = 178$ cells, $n = 129$ mGCs, $n = 49$ abGCs). See "Methods" section for details of bootstrapping procedure. **c** Ensemble activation matrix from one subject, plotted against event number, ordered in time ($K = 6$ ensembles, $M = 2473$ events). See Fig. S2 for cross-validation across different mice and model fits. **d** The spatial distribution (left) of ensemble activations from the six ensembles identified in C, colored by weights, color intensity representing the strength of each cell's association with an ensemble (A.U., right) (red = abGC, blue = mGC). See 4G and Fig. S2 for cross-validation across mice and model fits. **e** LER learned ensembles as a sort on the cell–cell correlation matrix (Kendall tau). Cells within a single ensemble have correlated activity patterns, while cells in different ensembles tend to have low correlation. **f** Decoding the most active ensemble from the LFP power spectrum during IEDs. The activation of each ensembles is associated with some "signature" identifiable in the LFP. **g** Fraction of abGC (red) vs. mGC-dominated (blue) ensembles, plotted as a function of $K$, defined as the proportion of ensembles in which either mGCs or abGCs are overrepresented beyond the 95th percentile of a random shuffle (dashed line; see "Methods" section) (data presented as mean value +/− 95% CI).

(>90%) mGC ensembles were identified regardless of $K$, whereas the purest abGC ensemble only contained 80% abGCs at $K = 20$ (Fig. S2F). This result further suggests that certain mGCs are coupled to abGCs; given evidence that newborn cells have a more active and independent role in the network while mature cells are more integrated into the feedforward pathway[57], one possible explanation of this observation is that the abGC ensemble feeds input to these mGCs, which in turn propagate this activity downstream. Together, we conclude that different epileptic animals organize ensembles with similar statistical properties, suggesting conserved structural and functional motifs underlying the epileptic network.

**Sharp-wave ripples recruit synchronized mixed ensembles of abGCs and mGCs.** To verify that these observations about ensemble structure are a feature of TLE rather than a physiological feature of the DG network in response to any electrophysiological event, we imaged DG while recording sharp-wave ripples (SPW-Rs) in stratum pyramidale of CA1 in non-epileptic mice (Fig. S4A). We then used our model to examine the ensemble structure underlying SPW-Rs in the control DG. As in Fig. 4, we compute a binarized "recruitment" matrix of cell activity that is significantly positively modulated within SPW-Rs compared to non-SPW-R epochs (see "Methods" section). From the recruitment matrices, we observe qualitatively that the abGC and mGC populations appear to be activated more sparsely and more synchronously with each other than in IEDs in TLE (Fig. S4B). As before, we train the model on the binarized recruitment matrix (Fig. S4C,D). We find that the non-epileptic DG does not segregate into abGC and mGC ensembles; instead, all inferred ensembles are mixed. Nonetheless, we do confirm that the learned ensembles correspond to real correlation structure in the underlying cells (Fig. S4E). To quantify this desynchronization across animals, we calculated Kendall's tau correlation between the median event responsiveness vectors of the two populations in each recording session, and find that abGCs are significantly less synchronous with mGCs in IEDs in TLE compared to in SPW-Rs in control ($p = 0.017$, Mann–Whitney $U$-test on $N_{TLE} = 17$ IED recording sessions from five mice, $N_{CTRL} = 8$ SPW-R recording sessions from two mice; Fig. S4F). Furthermore, to correlate this abGC-mGC desynchronization with chronic TLE, we looked at the seizure area-under-curve (AUC) measure of electrographic seizure severity over a 24-h vEEG period while mice were in their home cage. Regression lines fit to either IED synchrony or "event synchrony" (also including non-epileptic mice) vs. seizure AUC suggest an inverse relationship between abGC-mGC event synchrony and severity of the epileptic phenotype. The inverse relationship we find between

seizure AUC and abGC-mGC synchrony is consistent with our finding that the two populations are significantly desynchronized in IEDs in TLE compared to in SPW-Rs in the nonepileptic animal (Fig. S4G).

**Discussion**

How macroscale pathological events emerge from microscale changes at the cellular level has been a longstanding open question in the study of TLE[9]. Previous imaging and electrophysiological work examining the role of specific cell types in epileptic pathology has shown that inhibitory circuits play a critical role in shaping interictal dynamics in CA1[1,64,65]. However, relatively little is known about how the activity of the DG— the "gating" entry node to the tri-synaptic circuit—shapes downstream excitation–inhibition dynamics in vivo through excitatory output during these events. Changes within the DG abGC network have been hypothesized to contribute to the breakdown of the dentate gate which enables seizures to occur[25,26,30,34,38]. This question is clinically relevant, as any disease-modifying targeted therapy will depend on a robust understanding of the functional targets of the disease. In this study, we focused on the contribution of abGCs to epileptic DG circuits involved in IEDs and present an approach to evaluate the contribution of genetically identified neural populations to IEDs. We recorded abGCs and mGCs during IEDs, which reveal "snapshots" of discrete functional components underlying the epileptic network, and collated these snapshots under a generative model framework of ensemble recruitment to deduce the hidden functional organization within these networks.

The results presented here describe how populations of abGC and mGCs within the epileptic DG network differentially participate in macrolevel interictal events in vivo. Consistent with previous work in vitro[2,18,58], we find significant heterogeneity in LFP and calcium responses between IEDs, with distinctly identifiable pure mGC-driven events. The events in which a cell participates constitute a unique fingerprint that is informative about the cell's developmental lineage (abGC vs. mGC). We also find significant heterogeneity in the population responses of abGC and mGC cell-types to individual events, as well as in the within-population cell recruitment across cell-type dominated events.

We introduce Latent Ensemble Recruitment (LER), a biologically-motivated generative model of cell recruitment by interictal events. This model applies Bayesian topic modeling to in vivo two-photon calcium imaging data, and presents a conceptual framework for relating interictal events to ictogenic circuits: each microlevel population event represents a "snapshot" of a much wider macrolevel network, whose functional or synaptic

organization is not directly observable, yet whose microlevel calcium response at each time point is. Each one of these snapshots reveals some information about the underlying network, that is hypothesized to be structured on multiple scales to support seizure initiation and propagation[66,67]. While each individual event is relatively uninformative on its own, by accumulating these snapshots in an unsupervised way, it is possible to reconstruct a more complete picture of the underlying macro network structure. When fit to real data, this cell type-agnostic and event time-agnostic model discovers microensembles with clear cell-type and temporal organization. This observation suggests that, while the recruitment of any individual cell by IEDs may appear to be random, the recruitment of ensembles exhibits statistical regularities which strongly imply a role for the underlying microcircuits in the generation of epileptiform events. In particular, the regularities which we observe correspond to biological intuitions about the underlying circuits—e.g., the observed repeated activations in time may correspond to a microcircuit entering and remaining in a hyperexcited state for several seconds, such that the circuit is likely to be recruited by global events which occur within that window. Our modeling approach revealed that IED-associated variance in network activity in the epileptic DG is disproportionately correlated with the activity of abGC-dominated ensembles. We finally explored the implications of this model in the context of SPW-R related ensemble recruitment. We find that the abGC and mGC populations are more synchronous in SPW-Rs and organize into more sparsely activated mixed ensembles compared to IEDs. These observations suggest that within DG microcircuits, these two populations perform ripple-associated computations in conjunction; in contrast, the emergence of desynchronized, population-specific ensembles in TLE suggests that abGCs decoupling from mGCs and becoming coupled to each other may be one of the patho-mechanisms of TLE. These observations are consistent with the hypothesis that abGC-dominated ensembles drive pathological IED-associated network activity. However, the experiments performed here cannot definitively exclude alternative hypotheses, such as the possibility that IED and ensemble heterogeneity are driven by variability in the origin of IEDs; testing such hypotheses could form the basis of future experiments.

While our approach combines multipopulation two-photon calcium imaging and LFP recording in vivo, the difference in temporal resolution between LFP events (ms) and calcium signals (100 s of ms) is an important limitation, meaning unambiguous attribution of LFP events to calcium events is not always possible. We opted not to use spike deconvolution because spike deconvolution has not been validated for dentate granule cells, and there remain a number of unresolved questions surrounding the approach, especially on non-pyramidal populations[68]. To circumvent this problem, we average the calcium traces of many cells and IEDs in order to isolate the cellular IED response by the law of large numbers.

In this work, we investigated whether features of the low-dimensional global LFP signal can be decoded from the activities of individual cells and ensembles. The critical inverse problem of whether the (high-dimensional) population activity can be predicted from the one-dimensional global LFP signal is of great interest, though more difficult as it requires predicting a non-linear statistic of high-dimensional data from a one-dimensional LFP signal. Our finding that, for small ensemble numbers, the most active ensemble in each IED could be decoded above chance from the power spectrum of the IED alone suggests that specific GC ensembles do have distinct LFP signatures (see Valero et al.[69] for another example). The number of ensembles that can be decoded from this low-dimensional signal does not necessarily reflect the number of ensembles in the network; rather, the complexity of decoding increases combinatorially with the number of ensembles to be decoded. We believe decoding of larger ensemble numbers would improve with more training data, though this is outside the scope of this current work. While there is no obvious way to equate ensembles across animals, whether these LFP signatures are conserved across animals poses an interesting open question for future work. We hypothesize that there exists an equivalence relation between the LFP spectral signatures of IEDs (whether from the same or different animals). Combined with our observations here, this would imply that an equivalence class structure exists for cellular ensembles across animals via their LFP signatures, possibly offering an effective bridge to the micro–macro disconnect. While our results apply to the adult-born and mature subpopulations of the principal cell population in the DG, our framework of latent ensemble recruitment could motivate future experiments to also examine DG interneurons and mossy cells, which also play critical roles in the epileptic DG circuit[70–72], and more generally, to investigate cell type-specific ensemble dynamics in other HPC subregions during inter-ictal and ictal events[1,64,65].

Recent work[73] has reported that rAAVs impair neurogenesis in the adult mouse DG, with significant implications for functional recordings of GCs relying on rAAV for delivery of a genetically encode actuators or sensors. Given the design of our study, we believe these findings have limited relevance to the work presented here: principally, Johnston and colleagues[73] found that cells born two weeks or more prior to viral injection demonstrate no reduction, and in our study induction of Nestin expression by TMX occurred 2–3 weeks prior to injection with AAV1.Syn. GCaMP6f. In addition, the total volume of virus used in this study, 196 nl, is less than 1/5 the volume found to cause death of new-born cells at the same titer. Finally, even at the larger volume of 1000 nl, Johnston et al. only observed a "partial effect" of ablation at the viral titer used in this study, $1 \times 10^{12}$. Most compellingly, during imaging 3 weeks post injection, we empirically see a large and active population of abGCs indelibly labeled within the NestinCreER$^{T2}$ line—if this population has somehow been reduced, this effect would make our observation of an identifiable distinct role for abGCs in TLE even more remarkable.

While our method required ipsilateral partial aspiration of dorsal CA1 to gain high-resolution optical access to DG, normal DG-mediated neural dynamics has been demonstrated using similar methods in non-epileptic mice[57,74,75]. While normal levels of neurogenesis and general abGC morphology are not altered following aspiration[57], we cannot rule out the possibility that the damage caused by the implant contributed to the microcircuit reorganization and functional pathogenesis produced by the TLE model. This is an important question, and we expect that advances in three-photon imaging[76] will permit dissection of DG circuits with CA1 intact in the near future.

Although a rAAV-driven expression of GCaMP labels both excitatory and inhibitory populations in the DG, excitatory neurons form the overwhelming majority of the imaged population in this study. To quantify the potential impact of GABAergic neurons labeled among the DG GCs in the KA model, we imaged the GC layer in a Vgat-cre x Ai9-tdTomato mouse line, finding that GABAergic neurons only constitute 2.2% of the neurons in a typical FOV (Fig. S3). Therefore, while there may be a small number of unidentified interneurons in the extracted GC population, we assume that their impact on the results presented here is negligible.

Identification and characterization of circuit-level functional organization represents a critical step toward a general framework for resolving the micro-macro disconnect in chronic epilepsies[9]. Here we provide in vivo characterization of single cell and microcircuit-level dynamics in multiple cell populations in the

epileptic DG and relate them to macroscale events in simultaneously recorded LFP. These results may suggest a new design strategy for closed-loop systems for intervening in epilepsy, based on actively recognizing the patient-specific microcircuit targets for intervention in situ. Two-photon calcium imaging is currently the only technique that permits the longitudinal functional characterization of physiological and pathological neural circuits at the cellular level, which is necessary for the dissection of epileptic microcircuits and their recruitment during IEDs in vivo. Currently, continuous monitoring of microcircuit dynamics using calcium imaging is not translatable to humans. However, long-term scalp and invasive EEG recording is routinely conducted as part of the pre- and post-surgical evaluation for intractable focal epilepsy syndromes, and is the de facto diagnostic and monitoring tool for abnormal brain activity. The input signal to any therapeutic closed-loop intervention system will be electrographic for the foreseeable future, and the critical "inverse problem" for relating electrophysiological observables back to circuit mechanisms is unavoidable. Our work here takes the first steps toward solving this "inverse problem" by connecting signatures in LFP to circuit-level events, which will be essential for the design of new-generation functional closed-loop interventions.

## Methods

**Experimental model and subject details**. Animals: All experiments were conducted in accordance with the US National Institutes of Health guidelines and with the approval of the Columbia University Animal Care and Use committee. Male transgenic mice were obtained from The Jackson Laboratory to establish a local breeding colony on a C57BL/6J background: Nestin-CreER[T2] (JAX:016261) and ROSA26-CAG-stop[flox]-tdTomato Ai9 (JAX:007909). The Nestin-Cre line was crossed with the Ai9 reporter line to express tdTomato in adult-born granule cell populations. Mice were housed in the vivarium on a 12 h light/dark cycle (lights on at 07:00), constant temperature (21–24 °C) and humidity (30–50%), were housed 3–5 mice per cage, and had access food and water ad libitum. Mice were housed individually during video-EEG monitoring following kainic acid injection. Mature male and female mice (>8 weeks of age) were used for all experiments.

Induction of transgene expression and experimental timeline: Expression of the transgene in all mice was induced at approximately eight weeks of age, two weeks prior to kainic acid (KA) injection. Induction in Nestin-CreER[T2/tdTomato] mice, involved injection of 3 mg tamoxifen (TMX) (20 mg/mL in corn oil/10% ethanol) I.P./day for 5 consecutive days. abGCs were indelibly labeled with tdTomato following injections of TMX that drove expression of Cre in Nestin+ cells. This induction procedure labels approximately 90% of the immature granule cell population expressing doublecortin[48]. Two weeks later, KA was injected into the ventral hippocampus to induce the epilepsy model, and GCaMP6f injected into the dorsal dentate gyrus. Following recovery from injection of KA, mice were placed in video-EEG enabled housing where LFP and behavioral activity were continuous recorded to monitor ictogenesis[77]. Three weeks post-KA injection, mice were habituated to being head fixed under the two-photon microscope, and Ca$^{2+}$ imaging proceeded over the following 1–2 weeks (Fig. 1c).

Imaging window implant: Recombinant adeno-associated virus carrying the GCaMP6f gene (AAV2/1:hSyn-GCaMP6f) was obtained from Addgene (100837-AAV1) with titer ≥1 × 10$^{12}$. The dorsal dentate gyrus was stereotactically injected using a Nanoject syringe (Drummond Scientific) with a pulled glass capillary. Injection coordinates were −2.3 mm AP, 1.5 mm ML, and −1.8, −1.65, −1.5 mm DV relative to the cortical surface. Sixty-four nanoliter of virus was injected at each DV location in 32 nL increments at a flow rate of 23 nL/s. Three days later, mice were then surgically implanted with an imaging window (diameter: 2.0 mm; height: 2.3 mm) over the left dorsal dentate gyrus. Imaging cannulas were constructed by adhering a 2 mm glass coverslip (custom cut by Potomac) to a cylindrical stainless-steel cannula (2 mm diameter × 2.3 mm height) using optical adhesive (Norland). The imaging window was implanted 100–200 μm above the hippocampal fissure, providing optical access to the granule cells in the dorsal blade of the DG, and the interneurons and mossy cells in the hilus. Briefly, following induction of anesthesia (Isoflurane: 3.5% induction, 1.5–2.0% maintenance; 1.0 L/min O$_2$) and administration of analgesia (Metacam 5 mg/kg i.p.; Bupivacaine 2 mg/kg s.c.), the scalp was removed, and a 2.0 mm diameter craniotomy centered over the injection location was performed, using a fin-tipped dental drill. The dura was removed, and the underlying cortex aspirated until fibers within the *stratum lacunosum moleculare* were visible. The cannula with window was placed within the aspirated cavity and fixed to the skull with dental cement. A stainless steel straight headpost was then cemented to the skull posterior to the imaging window.

Electrode implants: During the window implant surgery, electrodes were implanted to monitor hippocampal local field potentials and neck muscle electromyographic signals. A custom monopolar electrode was constructed from

127 μm Teflon coated stainless-steel wire (A-M Systems), and inserted near the location of the KA injection in the ventral hippocampus (AP: −3.28 mm, ML: 2.75 mm, DV: −2.5 mm) ipsilateral to the imaging window to monitor and record hippocampal local field potentials. The location of the monopolar depth electrode was chosen based on previous studies showing that spontaneous seizures in the KA epilepsy model typically arise from the hippocampal formation ipsilateral to the KA injection site[78]. A stainless-steel jewelers screw was placed in the contralateral frontal bone for a ground electrode. To record electromyographic signals, a second stainless-steel screw was inserted in the occipital bone through overlying cervical trapezius muscle. Electrode wires were connected to a custom plug (Mill-Max strip connector) that was then cemented to the headpost. Following recovery from the implant procedure, mice underwent 24 h video-EEG monitoring for seizure detection. Following similar implant procedures, a subset of control mice was implanted with a custom bundled 4-channel electrode constructed from 51 μm PFA coated tungsten wire (A-M Systems) to monitor LFP sharp wave ripples. This probe was implanted in CA1 contralateral to the imaging window, and inserted at a 45° angle lateral to midline at the coordinates 2.3 mm AP, 2.75 mm ML relative to Bregma, and 0.9744 mm DV relative to the cortical surface.

Kainic acid injection: KA (AG Scientific, USA) was dissolved in sterile phosphate buffered saline with a final concentration of 20 mM. Using procedures described above, mice were stereotaxically injected while under isoflurane anesthesia, with 50 nL kainic acid unilaterally into a single location within the ventral hippocampus (AP: −3.28 mm, ML: 2.75 mm, DV: −3.4 mm). The location of the KA injection was chosen to allow for chronic 2-photon imaging of the dorsal dentate gyrus ipsilateral to the KA injection. Intraventral hippocampal KA injection results in epileptogenesis and seizure profiles similar to that found in the dorsal hippocampal KA model[55].

In vivo 2-photon imaging of dentate gyrus: All imaging was conducted using a two-photon microscope equipped with an 8 kHz resonant scanner (Bruker). Approximately 50–100 mW of laser power under the objective was used for excitation (Ti:Sapphire laser, (Chameleon Ultra II, Coherent) tuned to 920 nm), with adjustments in power levels to accommodate varying window clarity. To optimize light transmission, we adjusted the angle of the mouse's head using two goniometers (Edmund Optics, +/− 10-degree range) such that the imaging window was parallel to the objective. A piezoelectric crystal was coupled to the objective (Nikon 40X NIR water-immersion, 0.8 NA, 3.5 mm WD), allowing for rapid displacement of the imaging plane in the z-dimension. We continuously acquired red (tdTomato) and green (GCaMP6f) channels separated by an emission cube set (green, HQ525/70m-2p; red, HQ607/45m-2p; 575dcxr, Chroma Technology) at 512 × 512 pixels covering 225 μm × 225 μm, at 8–30 Hz (dependent on number of planes imaged) with photomultiplier tubes (green GCaMP fluorescence, GaAsP PMT, Hamamatsu 7422P-40; red tdTomato fluorescence, multi-alkali PMT, Hamamatsu R3896). A custom dual stage preamp (1.4 × 10$^5$ dB, Bruker) was used to amplify signals prior to digitization.

For all experiments, mice were head-fixed and ran freely on a 2 m long treadmill belt. We habituated the mice to the head-fixed condition for at least 1 h per day over three days prior to the beginning of the experiment.

Electrophysiology recording: All mice were implanted with a hippocampal LFP electrode and imaging window. During two-photon imaging, hippocampal LFP was recorded so that electrographic events could be correlated and analyzed with calcium imaging data. Electrophysiology signals were acquired with a multichannel digital recording system (Intan Technologies, USA) at 25 kHz and synchronized with the frame-start signal of the microscope. While not being imaged, mice were routinely monitored for interictal and seizure events using a custom continuous video-EEG comprising an analog multichannel recording system customized to record up to 16 mice simultaneously (NeuraLynx, USA). Briefly, the hippocampal LFP signal and video were acquired by a PC running a custom MATLAB (version R2011b) seizure recording and detection algorithm[77]. EEG signals were acquired for offline determination of IED, and seizure frequency and duration.

Ca$^{2+}$ imaging data preprocessing: All imaging data were pre-processed using the SIMA software package[79]. Motion correction was performed using whole frame registration. In cases where motion artifacts were not adequately corrected, the affected data were discarded from further analysis. We used the Suite2p software package[80] to identify spatial masks corresponding to neural regions of interest (ROIs) and extracted associated fluorescence signal within these spatial footprints, correcting for cross-ROI and neuropil contamination. Identified ROIs were curated post-hoc using the Suite2p graphical interface to exclude non-somatic components. ROI detection with Suite2p is inherently activity-dependent, and so for each session, we detected only a subset of neurons that were physically present in the FOV. To track ROIs across imaging sessions, all recordings of the same FOV were first concatenated before calculating spatial masks with Suite2p.

Running modulation calculation: For each calcium trace triggered on an event, the mean of the pre-event activity was subtracted from the mean of the post-event activity (mean $\Delta F/F_{post-event}$ − mean $\Delta F/F_{pre-event}$) to calculate the response magnitude. To calculate whether a cell reliably changes its activity on an event type, the response magnitudes for that event type were sampled with replacement, and the mean response magnitude was calculated. We repeated this calculation for 3000 bootstrap resampling iterations, to construct a 99% confidence interval (CI). The cell was determined to have be significantly negative response if the CI was less than 0, significantly positive response if the CI was greater than 0, and non-significant response if the CI contained 0.

Detection of sharp wave ripple events: LFP signals were calculated from the wide-band 25 Hz signal by down-sampling to 1250 Hz. From the four channels on the probe, the wire within the pyramidal layer was identified as showing the greatest ripple-band Gabor wavelet power (100–225 Hz). After the removal of noisy LFP epochs, SPW-R events were detected using a custom supervised algorithm based on a template hand-labeled ground-truth data set of SPW-R events from four mice (separate from those recorded in this study) using k-nearest-neighbor embedding based on wavelet-derived SPW-R features (Matlab version R2019a). Candidate events identified using the supervised template matching procedure, were considered SPW-R-events if their within-event ripple-band wavelet power was at least 12 median absolute deviations above the median for the session. SPW-R-event detection was visually verified. Only SPW-R-events occurring during periods of immobility lasting at least 3 s were included in the analysis. For calculations requiring point time estimates, the within-SPW-R ripple-power peak was used.

Classification of cell type and epileptiform events: IEDs were determined from single-channel LFP using a semi-supervised approach. First, the LFP signal was aligned with each imaging frame using an external frame-trigger ADC channel. The magnitude spectrum of the LFP snippet corresponding to each imaging frame was calculated using the Fast Fourier Transform (FFT). The magnitude spectrum was chosen so that IEDs could be detected regardless of phase within the frame. We then trained an Online Kernelized Perceptron (using a Gaussian radial basis function kernel with $\sigma = 1$) to recognize frames containing an IED by hand-annotating a small number of LFP snippets containing IEDs, allowing the classifier to classify all the snippets in the session based on the hand-labeled examples, and updating annotations based on the classifier's feedback. This classification procedure was conducted according to an iterative feedback process, where human input would train the algorithm while correcting the algorithm's mistakes, so that each session is hand-curated, but the amount of hand curation necessary is minimized. We verified the internal consistency of this algorithm quantitatively by plotting the $1/f$ corrected average IED power spectrum (Fig. 2b); we find the classified IEDs share common spectral characteristics. An online perceptron-type algorithm was chosen because it permits efficient human-supervised training with immediate feedback after each labeled example and accurate phase-agnostic classification with a small number of labeled samples. A kernelized algorithm was chosen so that nonlinear relationships between IEDs and the magnitude spectrum could be captured.

Responsiveness of cells to IEDs: A scalar responsiveness $\Delta$ of a cell to an event was calculated as the cell's mean activity in a fixed 3-s window postevent minus the cell's mean activity in the same window pre-event. An $N_{cells} \times N_{events}$ data matrix was constructed in this manner, and a $N_{cells} \times 1$ vector of labels was constructed from the red channel, with tdTomato-tagged cells labeled as "abGC" and all others labeled as "mGC".

Linear classification of abGCs and mGCs based on IED response profiles: A logistic regression linear classifier was trained to classify abGCs and mGCs based on their IED response profiles. Training and testing were performed using the scikit-learn package version 0.23.1[63].

To assess the data for linear separability and examine the learned weights, logistic regression was first trained on the entire dataset using an L2 regularization penalty to avoid overfitting (Fig. 3a, b). To evaluate the generalization of the model, the dataset was split by cell into 80–20 training and test sets, and test accuracies were computed. The model was trained with L2 regularization to establish an accuracy baseline to compare with L1 accuracy.

UMAP dimensionality reduction: We use supervised UMAP to perform dimensionality reduction in order to visualize the cell IED response profiles in high dimensional space. Dimensionality reduction was performed using the UMAP-learn package[81,82]. The UMAP embedding was cross-validated in a similar manner to the logistic regression: UMAP was trained on a random 80% of the cells and the learned embedding was applied to a held-out on 20% of the cells.

Latent ensemble recruitment modeling: Synchronous network events were identified using spectral analysis of a simultaneously collected single-channel LFP recording in ipsilateral CA1. The scalar responsiveness $\Delta$ of a cell to an event is defined as above. The responsiveness was binarized to an "activation" variable $z'$, using the following bootstrapping procedure: A bootstrap responsiveness distribution was calculated by drawing 10,000 random trigger frames with replacement from the $\Delta F/F$ trace of each cell, excluding IEDs and seizures, and then used to construct a 95% confidence interval. A cell was determined to be "activated" by an IED if its responsiveness to that IED exceeded the 95th percentile of the bootstrapped responsiveness distribution. Existing data from in vitro physiological studies[1] have suggested that epileptiform activity is associated with spatially clustered, temporally coactivated neural ensembles. Our model (Fig. 4a, b) formalizes a weaker version of this hypothesis: that each interictal spike recruits a sparse set of pre-existing ensembles within the network, and each ensemble activates a subset of its constituent cells.

Specifically, we assume the following generative process for a dataset consisting of $M$ events in a population of $N$ cells organized into $K$ ensembles:

(1) Once for the entire network, we sample the ensembles $\sigma_k$ for each $k$ from a Dirichlet distribution parameterized by sparsity-controlling hyperparameter $\beta$. This gives us $K$ ensembles, where each ensemble $\sigma_k$ is a probability vector over the $N$ cells in the network, whose entries correspond to the strength of each cell's association with that ensemble.

(2) For each event $i = 1, \ldots, M$, we sample $\vartheta_i$, a sparse probability vector over ensembles, from another Dirichlet distribution parameterized by hyper-parameter $\alpha$, representing the degree to which each ensemble is recruited by that event.

(3) For each cell,

  a. We sample the indicator variable $z_{ij}$ from a Bernoulli distribution parameterized by $p = \sum_k \theta_{ik} \sigma_{kj}$, which represents whether or not that cell was recruited to that event.

  b. We simultaneously sample an ensemble $\zeta$ from the multinomial Multinomial($\varphi$), which gives the ensemble from which the cell $j$ was recruited.

  c. Finally, we draw our observed responsiveness $\Delta$ from a Gaussian, either $N(\mu_{on}, \sigma_{on}^2)$ or $N(\mu_{off}, \sigma_{off}^2)$ depending on the value of $z_{ij}$.

Ensembles dominated by one population: We identified "abGC dominated" and "mGC dominated" ensembles by sampling from a shuffle distribution as follows: The number and sizes of ensembles were fixed from the ensembles identified in real data. The cell type labels were then randomly permuted while preserving the overall number of abGCs and mGCs in the entire population. This procedure was repeated for 10,000 iterations to construct null distributions for the number of abGCs and mGCs expected in each ensemble based on its size. Ensembles which contained more abGCs or mGCs than the 95th percentile of its size-matched shuffle ensemble were termed abGC-"dominated" or mGC-"dominated", respectively.

Inverse classification: We trained multiple iterations of the model, varying the $K$ hyperparameter from 2 to 10. We then decoded the most active ensemble from the LFP power spectrum of the imaging frame during which the IED occurred using a Random Forest classifier (from the scikit-learn package) with 100 decision tree estimators trained on the LFP power spectrum of that frame, validated across 10 random 80–20 train-test splits.

Synchrony and seizure AUC: The "synchrony" of abGCs and mGCs is defined as the Kendall tau correlation between the median event-activation vector of each population. Kendall's tau was chosen as a "soft" correlation able to tolerate small displacements in time, meaning desynchronization in time is penalized on a continuous basis rather than as an all-or-nothing correlation as with e.g., Pearson's r.

**Reporting Summary**. Further information on research design is available in the Nature Research Reporting Summary linked to this article.

## Data availability

All data supporting the findings of this study are provided within the paper and its supplementary information. All additional information will be made available upon reasonable request to the authors. Source data are provided with this paper.

## Code availability

The data analysis for this study was mainly performed using Python 2.7 and the open source packages scikit-learn and SIMA, which are available at https://github.com/scikit-learn/scikit-learn and https://github.com/losonczylab/sima/. Custom code used in this study is also available at https://github.com/losonczylab (https://doi.org/10.5281/zenodo.4158960 and https://doi.org/10.3389/fninf.2014.00077).

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

## Acknowledgements

Supported by grants NINDS-1U19NS104590 (A.L., I.S.), NINDS-1R01NS094668 (A.L., I.S.), Kavli Foundation (A.L.), American Epilepsy Society Junior Investigator Award (F.T.S.), NINDS-1F31NS120783-01 (Z.L.) and T32GM007367 (Z.L.). We would like to thank P. Maccario for help running IED classification, H. Kim for assisting with vEEG software, and D. Hadjiabadi and C. Schevon for comments on the manuscript.

## Author contributions

F.T.S., Z.L., I.S., and A.L. designed the experiments. F.T.S. and Z.L. collected and analyzed the data. F.T.S. and W.L. designed electrophysiology hardware components. F.T.S., Z.L., and A.G. collected and analyzed control SPW-R recordings. F.T.S., Z.L., I.S., and A.L. wrote the manuscript.

## Competing interests

The authors declare no competing interests.
