## [Peer Review File · Nature Communications]

Reviewers' Comments:

Reviewer #1:

Remarks to the Author:

The study by Sparks and colleagues aims understanding the network mechanisms, which may give rise to interictal epileptiform discharges (IEDs) in the mouse dentate gyrus. By applying a genetic labeling method which allows the differentiation of adult-born and mature granule cells (abGCs, mGCs, respectively) and chronic 2-Photon imaging of the neuronal population in the dentate gyrus, they obtain activity of abGCs and mGCs. Moreover, they developed a generative model motivated by Bayesian topic models to make predictions on the ensemble structure and GC identities underlying the IEDs. They conclude from the model that

- (1) IEDs are driven by multiple ensembles with different cellular components (mGCs, abGCs) and different ensembles are associated with different LFPs with distinct signal structure.
- (2) abGCs dominate ensembles despite their numeric minority in the dentate gyrus network.
- (3) some mGCs are coupled to abGCs and somehow seem to propagate mGCs' activity within the network.
- (4) epileptic animals organize ensembles with quite similar statistical properties.
- (5) the reorganization of the network upon temporal lobe epilepsy produces functionally distinct ensembles of GCs with specific pathophysiological characteristics.

The manuscript is very well written and very clear in its messages and modeling procedures. It nicely combines in vivo characterization of neuronal activity with functional organization of the microcircuitry and macrocircuit LFP signals. I have therefore only few comments.

Major comments:

One of the main findings is that abGCs are significantly more responsive to the IED than mGCs and refer in this context to figure S1A, in which z-scores of responsiveness are plotted. It would be of general interest to mention mean activities as dF/F for both populations to be able to compare them with WT controls. The responsiveness is close to '0' in both populations, which makes it hard to understand how a significant difference of $p < 0.001$ can emerge, particularly when only 5 mice were tested per group. Moreover, since only mild changes in the activity were observed (pre- vs post IED), how reliable are the detected differences in the responsiveness in dependence on thresholding of signal detection?

From the surgical procedure (aspiration until stratum lacunosum moleculare in Methods and reference to Danielson et al., 2016a) it remains unclear, whether CA1 was damaged in the here imaged mice. If so, experiments should be confirmed in mice with CA1 intact.

Minor comments:

Figure 1: Please mention whether 1B refers to single frame image or to an average image. Include in (D) the label 'LFP' to the LFP trace and include arrows pointing to IEDs.

Reviewer #2:

Remarks to the Author:

Sparks and Liao et al. investigate the ensemble activity structure of the dentate gyrus (DG) in an epileptic mouse model. By crossing different transgenic mouse lines, the authors are able to genetically identify adult-born granule cells (abGCs) in the DG in-vivo. Combining 2-photon calcium imaging, local field potential recordings, and a model of chronic temporal lobe epilepsy, the authors record the activity of both abGCs and mature GCs (mGCs) during interictal

epileptiform discharges (IEDs). They use statistical models to analyze the involvement of different populations during IEDs and find that the more abGCs-matching ensembles are more prominently recruited by IEDs. The topic of the research is interesting in that it studies the contribution of distinct populations of granule cells in the DG to epilepsy, of which the cellular level mechanisms are unclear; the specific roles of abGCs are important to understand but these have been difficult to dissect previously due to lack of proper tools. However, the analysis is preliminary and there are several major issues with the paper as is, mostly pertaining to the interpretation of the results, statistical analysis, and missing critical details. Please find below detailed comments:

1. AAV2/1:hSyn-GCaMP6f labels not only granule cells but also interneurons in the DG. They would be difficult to be differentiated from cellular calcium signals but may be involved in distinct network activity thus may form distinct ensembles; the ensemble structure could be picked up by the statistical analysis shown Fig. 4. The authors would need to discuss how this may influence the interpretation of Fig. 4. More specifically, is it possible that one of the major ensembles shown in Fig. 4C is mainly comprised of interneurons? This would have important implications on the overall interpretation of the DG GC network (instead of the entire network) during IEDs. Related to this, the labelled abGCs are >5 weeks old at the time of data acquisition, is it possible that the more newly-born GCs actually do not express tdTomato but GCaMP6f thus contaminate the 'mGCs' population?

2. Fig. S1A: the correct test to use is Wilcoxon signed-rank test because it's the same mice and should not be treated as two independent samples.

3. Does the average spectrum shown in Fig. 2B correspond to only the interictal spikes as the ones shown in Fig. 2A upper right? What do the error bars indicate? Why does the spectrum peak around 250-400 Hz (does not seem to correspond to the example shown in Fig. 2A upper right)?

4. In Fig. 3, the logistic regression without regularization is not blind to cell identity because the weight of each individual IED is assigned based on its recruitment of either more mGCs or more abGCs (Fig. 3A&B). The regression with L1 regularization should be blind to cell identity, is that the case? Otherwise, the analysis & interpretation is circular. The authors also use L2 regularization penalty (Fig. 3C) but the results are not mentioned in the main text. Are raw calcium time courses used throughout the paper? Is it possible that the ramping shown in dF/F in Fig. 3D is an artifact of the long time scale of calcium signals? Specifically, using deconvolved dF/F would be more rational to show the more precise time-locked transient responses. Also, plotting the PSTH for a time window of [-5 5] s seems too large considering that the IED rate is about 1.13/s. Moreover, the detailed methods on the linear classifier are completely missing from the Methods section.

5. In the caption of Fig. S2D: it's written "suggesting that certain mGCs are highly coupled to abGCs and may propagate abGC activity through the network". What is the logic in saying that? Please elaborate.

6. The correlation matrices shown in Fig. S2E are problematic. The correlations are based on the fraction of abGCs in each ensemble and these fractions are sorted from low to high. The correlation will be inherently high by construction once you sort them and then obtain the correlation coefficient. Therefore, the main claim of this figure that "the statistics of this ensemble structure are highly conserved across animals" does not hold.

7. Terminology in Fig. 3C caption: 100-fold cross-validation on an 80-20 train-test split. Do the authors mean "100 times 5-fold cross-validation"? 100-fold would imply splitting data into 99% for training and 1% for test.

8. The mice are head-fixed and free to run on a treadmill belt. How is the animal's behavior related to the heterogeneity in the responses of abGCs and mGCs to individual IEDs? It is possible that certain degree of the observed heterogeneity arises from different behavioral states. For

example, the responses of some abGCs or mGCs to IEDs may depend on if the animal is running or still, how fast the animals is running, etc. This is corroborated by that the most active ensemble can be decoded from the IED power spectrum (both could be influenced by animal's behavior) for small ensemble number. Unfortunately, this aspect is not analyzed at all. This is minor but the authors also implant an EMG electrode in the mice. How the EMG signal, which may reflect the animal's behavioral state, may explain the heterogeneity is not analyzed. It is also useful to check the relationship between the EMG signals and IEDs; is it possible that some of the IEDs and certain ensembles are correlated with EMG fluctuations? These control analyses would be helpful to exclude possibilities which could otherwise undermine the main conclusions.

9. In Table S1, please double check if the numbers are correct. The "Mean Total" seems wrong.

10. It is problematic that several of the conclusions are made on statistics from only one example animal (e.g. Fig. 2C, Fig. 3D) without analyzing the population statistics (all mice).

11. Although the ratio between abGC-dominated and mGC-dominated ensembles are $\sim 4:6$ and the percentage of abGCs is only $\sim 11\%$ in the DG, this does not necessarily imply that the abGC-dominated ensembles contribute to the generation of IEDs more than mGC ensembles. It is possible that abGCs are simply more heterogeneous (therefore more distinguishable ensembles), or the majority of IEDs-related mGCs occupy a small number of ensembles; consequently, the total fraction of neurons involved in IEDs could be higher in mGCs than abGCs, disproving that abGC ensembles drive IEDs more prominently than mGC ensembles. Also, without further experiments, one could only say how GCs activity is correlated with but not how they could 'drive' IEDs.

12. How is the bootstrapped, binarized matrix z' constructed? Does it solely depend on other synchronous network activity excluding IEDs? This seems to be the case as in the caption of Fig. 4B. Then the ensemble structure is estimated from this matrix and the results are used for the analysis of IEDs events. Does it not depend heavily on the hypothesis that the same ensemble structure is preserved across other synchronous network activity and IEDs?

Reviewer #3:

Remarks to the Author:

The manuscript by Sparks et al., describes the relative contribution of the activity of adult born dentate granule cells and mature granule cells to interictal epileptiform discharges in the hippocampus. The authors use several novel techniques to classify which populations of cells are activated in response to any given IED, and compare ensemble activity across animals. The authors find a set of latent ensembles of DGs involved in IED activity and these ensembles are disproportionately dominated by activity of abGCs. Comments are below:

1. My biggest concern with the manuscript is that it is unclear what these results mean for our understanding of epilepsy in general. For the composition of these ensembles to relate to epilepsy more broadly, it would be good to show that some result correlates with an epileptic phenotype. For example, the number of IEDs with pure abGC ensembles or the pureness of the ensembles in any given animal would relate to seizure frequency or duration on an animal-by-animal basis, or perhaps more tractably the ensemble composition could relate to the ability of the IED to generalize on an IED-by-IED basis. The assertion that the findings are an epilepsy-specific phenomenon and is not a function of how DG neurons behave in ensembles more broadly is also questionable. I do not see any effort made to distinguish DG ensemble activity in IEDs from activity in normal ongoing function or evoked stimulation. However, the technique used to characterize latent ensemble recruitment is interesting and novel.

2. Have the authors used the classifier on snips of control LFP to see if it catches sharp waves?

Can the authors use the movement information to inform whether they are capturing IEDs vs sharp waves?

3. The authors should include a figure showing the LFP classifier feature space.
4. How are the authors binning to classify a cell as responsive or non-responsive to an IED event, e.g. what is the time window for calcium changes in a cell in relation to an identified LFP IED that would classify that cell as "responsive"? How was this chosen? Over what period of time was activity averaged? Have the authors done a robustness analysis to validate their binning choice?
5. How are you determining the 0.1 cutoff for the IED mGC-ness score? What proportion of the IEDs have a sameness score between -0.1 and +0.1 (or is the histogram in 3B all the data from that animal)? Does the bi-modality hold for all animals?
6. The L1 regularization analysis seems tautological and difficult to cross validate. Can the authors show a scatterplot of responsive cells to each spike by cell type?
7. Is there some feature of the IED that can be mapped on to whether the ensemble is a mCG-/abGC-selective or a mixed ensemble?
8. Are data in figure 4A-F for one recording session in one animal? Do these results hold up across animals?
9. 4E, left is never described in the text
10. What's the cut off for an abGC- or mGC- "dominated" ensemble? Does this vary across animals? Does it vary as a function of the frequency of IEDs even within animal?
11. Are data displayed in 4H from all IEDs in all animals? Most example traces show 4 or 6 ensembles and text states that $K=6$ is validated across animals, but the accuracy of decoding the IED is at chance for those ensemble sizes. Could the authors please discuss?

Minor

All animals are presumably recorded for multiple test sessions but overall, the reader has to work pretty hard to determine which figures are from one recording vs one animal vs pooled over the entire cohort. It is unclear why more pooled data are not shown, unless the results do not generalize?

Red and blue are not labeled in Figure 3D

The organization on Figure 3 makes it hard to read

4H is cited before 4G

4G orange and blue are not labeled

Figures 4A, B, D and S2A are so small, they're almost impossible to see

Was the location of the LFP recording electrode in the CA1 cell layer verified?

We thank all our Reviewers for their enthusiasm for our work and for their constructive comments. Suggestions from the Reviewers greatly helped to improve the manuscript. Our point-by-point responses to their specific concerns are noted below with the Reviewers' comments retained in blue.

Reviewer #1 (Remarks to the Author):

The manuscript is very well written and very clear in its messages and modeling procedures. It nicely combines *in vivo* characterization of neuronal activity with functional organization of the microcircuitry and macrocircuit LFP signals. I have therefore only few comments.

We would like to thank the Reviewer for the positive commendation of our manuscript.

Major comments:

One of the main findings is that abGCs are significantly more responsive to the IED than mGCs and refer in this context to figure S1A, in which z-scores of responsiveness are plotted. It would be of general interest to mention mean activities as $\Delta F/F$ for both populations to be able to compare them with WT controls. The responsiveness is close to '0' in both populations, which makes it hard to understand how a significant difference of $p < 0.001$ can emerge, particularly when only 5 mice were tested per group. Moreover, since only mild changes in the activity were observed (pre- vs post IED), how reliable are the detected differences in the responsiveness in dependence on thresholding of signal detection?

Thank you for these insightful comments. The mean responsiveness is close to 0 as we expect, as IED recruitment is sparse in cells and in time (i.e., each IED only recruits a few cells, if any, and each cell is only recruited by a few IEDs, if any). The smaller p value occurs if the two groups of cells are pooled and tested against each other. We have revised this to more appropriately use a Wilcoxon signed-rank test on population means paired by mouse instead, which still yields a significant difference of $p < 0.05$.

In addition, we have clarified the thresholding procedure for binarizing cell responsiveness in Methods (page 29)

"The responsiveness was binarized to an "activation" variable z' , using the following bootstrapping procedure: a bootstrap responsiveness distribution was calculated by drawing 10,000 random trigger frames with replacement from the $\Delta F/F$ trace of each cell, excluding IEDs and seizures, and then used to construct a 95% confidence interval. A cell was determined to be "activated" by an IED if its responsiveness to that IED exceeded the 95th percentile of the bootstrapped responsiveness distribution."

From the surgical procedure (aspiration until stratum lacunosum moleculare in Methods and reference to Danielson et al., 2016a) it remains unclear, whether CA1 was damaged in the here imaged mice. If so, experiments should be confirmed in mice with CA1 intact.

Thank you for valuable input. The aspiration into CA1 is indeed a limitation of all existing approaches to cellular resolution two-photon imaging of the dentate gyrus. Unfortunately, while some preliminary evidence suggests that it may be possible in the future to image DG without aspiration using 3-photon calcium imaging, there currently do not exist standard, published techniques which allow the DG to be imaged without aspirating CA1.

We have added the following to the discussion (page 16).

“While our method required ipsilateral partial aspiration of dorsal CA1 to gain high-resolution optical access to DG, normal DG neural dynamics has been demonstrated using similar methods in non-epileptic mice (Danielson et al., 2016, Hainmueller and Bartos, 2018, Woods et al., 2020). While normal levels of neurogenesis and general abGC morphology is not altered following aspiration (Danielson et al., 2016), we cannot rule out the possibility that the damage caused by the implant contributed to the microcircuit reorganization and functional pathogenesis produced by the TLE model. This is an important question, and we expect that advances in three-photon (Weisenburger et al., 2019) imaging will permit dissection of DG circuits with CA1 intact in the near future.”

Minor comments:

Figure 1: Please mention whether 1B refers to single frame image or to an average image. Include in (D) the label ‘LFP’ to the LFP trace and include arrows pointing to IEDs.

Thank you. We have added this information to **Figure 1** and caption.

Reviewer #2 (Remarks to the Author):

Sparks and Liao et al. investigate the ensemble activity structure of the dentate gyrus (DG) in an epileptic mouse model. By crossing different transgenic mouse lines, the authors are able to genetically identify adult-born granule cells (abGCs) in the DG in-vivo. Combining 2-photon calcium imaging, local field potential recordings, and a model of chronic temporal lobe epilepsy, the authors record the activity of both abGCs and mature GCs (mGCs) during interictal epileptiform discharges (IEDs). They use statistical models to analyze the involvement of different populations during IEDs and find that the more abGCs-matching ensembles are more prominently recruited by IEDs. The topic of the research is interesting in that it studies the contribution of distinct populations of granule cells in the DG to epilepsy, of which the cellular level mechanisms are unclear; the specific roles of abGCs are important to understand but these have been difficult to dissect previously due to lack of proper tools. However, the analysis is preliminary and there are several major issues with the paper as is, mostly pertaining to the interpretation of the results, statistical analysis, and missing critical details. Please find below detailed comments:

1. AAV2/1:hSyn-GCaMP6f labels not only granule cells but also interneurons in the DG. They would be difficult to be differentiated from cellular calcium signals but may be involved in distinct network activity thus may form distinct ensembles; the ensemble structure could be picked up by the statistical analysis shown Fig. 4. The authors would need to discuss how this may influence the interpretation of Fig. 4. More specifically, is it possible that one of the major ensembles shown in Fig. 4C is mainly comprised of interneurons? This would have important implications on the overall interpretation of the DG GC network (instead of the entire network) during IEDs.

Thank you for the valuable input. It is true that GCaMP expression via synapsin promoter-driven rAAV labels all populations of DG neurons. Given the low abundance of interneurons compared to granule cells, it is unlikely that interneurons would significantly contaminate the mature granule cell population (note that abGCs are positively identified by the Nestin-Cre x tdTomato approach). In order to assess the potential impact of interneurons in DG mixed in with mGCs, we now performed a control experiment in which we imaged the GCL in a VGAT-cre x Ai9 mouse (interneurons express tdTomato) that was injected with KA. The time average field-of-view shown below (**Rebuttal Figure 1**, interneurons are in red), illustrating that while a few interneurons are present, they form only a small proportion of the cells recorded. Therefore, we do not anticipate that the analysis was meaningfully affected by the presence of interneurons. We have added the

following discussion (page 17) as well as an additional Supplementary Figure:

Rebuttal Figure 1 (Figure S3).

“Although a rAAV-driven expression of GCaMP labels both excitatory and inhibitory populations in the DG, excitatory neurons form the overwhelming majority of the imaged population in this study. To quantify the potential impact of GABAergic neurons labeled among the DG granule cells in the KA model, we imaged GCL in a VGAT-cre x Ai9-tdTomato mouse line, finding that GABAergic neurons only constitute a small fraction (~2%) of the neurons in a typical FOV (Figure S3). Therefore, while there may be a small number of unidentified interneurons in the extracted GC population, we assume that their impact on the results presented here is negligible.”

Related to this, the labelled abGCs are >5 weeks old at the time of data acquisition, is it possible that the more newly-born GCs actually do not express tdTomato but GCaMP6f thus contaminate the ‘mGCs’ population?

Thank you for this valuable comment. We have previously characterized the labeling efficacy of the transgenic method (Danielson et al., 2016) based on NestinCreERT2 mice were crossed with a conditional tdTomato reporter line and pulsed with tamoxifen. We found that labeling efficacy of the method is high: approximately 90% of the doublecortin population of immature GCs are labeled, indicating 10% of abGCs were not labeled and thus were not characterized as abGCs. Because of the irreversible nature of Cre-recombination in stem cell, we expect that the labeling is permanent and indelible after induction in the lineage.

2. Fig. S1A: the correct test to use is Wilcoxon signed-rank test because it’s the same mice and should not be treated as two independent samples.

Thank you, it is fixed.

3. Does the average spectrum shown in Fig. 2B correspond to only the interictal spikes as the ones shown in Fig. 2A upper right? What do the error bars indicate? Why does the spectrum peak around 250-400 Hz (does not seem to correspond to the example shown in Fig. 2A upper right)?

Thank you for the comment. We have revised the figure legend. The average spectrum corresponds to the average of all the interictal spikes recorded from this animal, with error bars showing the SEM. The peak is likely to reflect the timescale of positive and negative deflections (on the order of 10 ms in an event lasting ~50 ms), including multiple deflections in the case of multiphasic IEDs. In the raw magnitude spectrum, this peak is very small, and is emphasized by the $1/f$ correction.

4. In Fig. 3, the logistic regression without regularization is not blind to cell identity because the weight of each individual IED is assigned based on its recruitment of either more mGCs or more abGCs (Fig. 3A&B). The regression with L1 regularization should be blind to cell identity, is that the case? Otherwise, the analysis & interpretation is circular.

Thank you for the comment. If we understand the Reviewer's comment correctly, the Reviewer is concerned that the regression depends on the cell identity indirectly through the biases of the IEDs. We used logistic regression to ask the question of whether abGCs vs mGCs exhibited different IED response profiles - it is possible (and indeed we demonstrate) that e.g. mGC identity can be decoded from participation in events that other mGCs are known to participate in. The L1 regularization sharpens this argument by selecting for those events which are highly predictive. In this case, there is dependence on cell identity, but it is indirect and decoded via the IED responses - which was the goal of the analysis.

We have reorganized this section to emphasize this point:

We use L1 and L2 regularization of the classifier in order to further probe how cell identity influences participation in IEDs. The regularized classifier could find that participation in a particular sequence of mixed events is predictive of abGC identity, which would indicate that abGCs and mGCs are internally coherent and respond to IEDs through population-specific patterns. A second possibility is that the regularized classifier may find individual "pure" or near-pure mGC-associated IEDs, which would indicate that some IEDs are attributable almost exclusively to one population. Finally, a third possibility is that the regularization penalty results in a much worse test accuracy, which would indicate that the entire profile of a cell's response to IEDs is necessary to classify the cells accurately.

*Based on the cells' IED responses, an L2-regularized logistic regression classifier achieved $85\pm 5\%$ accuracy on a test set, balanced to remove the effect of population size, cross-validated over 100 random 80-20 train-test splits (**Figure 3C**, left; **Figure S1E**). Subsequently, we use a sparsifying (L1) regularization penalty in order to concurrently identify the subset of IEDs which are "most informative" about cell identity, in the sense that knowing a cell's response to those events is almost as predictive of identity as knowing the cell's complete response profile (**Figure 3C** right; see **Methods**). The L1 regularization penalty resulted in a slight reduction in test set accuracy ($81\pm 5\%$ accuracy using the same procedure), but identified IEDs with highly population-specific recruitment profiles and disregarded the majority which exhibited mixed activation, suggesting the second hypothesis above holds true, i.e., cell identity can be decoded from*

participation in events that other cells of the same type are known to participate in (**Figure 3C**; see **Methods**).

The authors also use L2 regularization penalty (Fig. 3C) but the results are not mentioned in the main text.

Thank you for the comment. Discussion of the L2 penalty has been added (page 9):

Based on the cells' IED responses, an L2-regularized logistic regression classifier achieved 85±5% accuracy on a test set, balanced to remove the effect of population size, cross-validated over 100 random 80-20 train-test splits

The following clarification has also been added to Methods (page 29):

To assess the data for linear separability and examine the learned weights, logistic regression was first trained on the entire dataset using an L2 regularization penalty to avoid overfitting (Fig 3A and B). To evaluate the generalization of the model, the dataset was split by cell into 80-20 training and test sets, and test accuracies were computed. The model was trained with L2 regularization to establish an accuracy baseline to compare with L1 accuracy.

Are raw calcium time courses used throughout the paper? Is it possible that the ramping shown in dF/F in Fig. 3D is an artifact of the long time scale of calcium signals? Specifically, using deconvolved dF/F would be more rational to show the more precise time-locked transient responses. Also, plotting the PSTH for a time window of [-5 5] s seems too large considering that the IED rate is about 1.13/s.

Thank you. DF/Fs are used throughout the paper. We have added the following Discussion:

While our work is, to our knowledge, the first to combine multipopulation two-photon calcium imaging and LFP recording in vivo, the difference in temporal resolution between LFP events (ms) and calcium signals (100s of ms) is an important limitation, meaning unambiguous attribution of LFP events to calcium events is not always possible. We opted not to use spike deconvolution because spike deconvolution has not been validated for dentate granule cells, and there remain a number of unresolved questions surrounding the approach, especially on non-pyramidal populations (Evans et al., 2019). To circumvent this problem, we average the calcium traces of many cells and IEDs in order to isolate the cellular IED response by the law of large numbers.

Moreover, the detailed methods on the linear classifier are completely missing from the Methods section.

We apologize for the omission. This has been now added (page 29):

Responsiveness of cells to IEDs

A scalar responsiveness Δ of a cell to an event was calculated as the cell's mean activity in a fixed window post-event minus the cell's mean activity in the same window pre-event. The analyses shown use a window size of 3 seconds, though the results do not change appreciably for window sizes between 1 second and 5 seconds (data not shown). An $N_{\text{cells}} \times N_{\text{events}}$ data matrix was constructed in this manner, and a $N_{\text{cells}} \times 1$ vector of labels was constructed from the red channel, with tdTomato-tagged cells labeled as "abGC" and all others labeled as "mGC".

Linear classification of abGCs and mGCs based on IED response profiles

A logistic regression linear classifier was trained to classify abGCs and mGCs based on their IED response profiles. Training and testing were performed using the scikit-learn package version 0.23.1 (Pedregosa et al., 2011).

To assess the data for linear separability and examine the learned weights, logistic regression was first trained on the entire dataset using an L2 regularization penalty to avoid overfitting (Fig 3A and B). To evaluate the generalization of the model, the dataset was split by cell into 80-20 training and test sets, and test accuracies were computed. The model was trained with L2 regularization to establish an accuracy baseline to compare with L1 accuracy.

5. In the caption of Fig. S2D: it's written "suggesting that certain mGCs are highly coupled to abGCs and may propagate abGC activity through the network". What is the logic in saying that? Please elaborate.

Thank you for this comment. We find certain mGCs which seem coupled to abGC ensembles. It is surprising that the "purest" abGC ensemble still contains a fair number of mGCs - if we believe that newborn cells have a more active and independent role in the network, as shown by us and others (Danielson et al., 2016, Sahay et al., 2011, Ikrar et al., 2013, Nakashiba et al., 2012), then one possible explanation of this phenomenon is that this abGC ensemble is feeding input to these mGCs, which, being more integrated into the circuit, propagates this activity downstream.

We have reworded this statement and moved it to the Discussion section (page 14).

Our modeling approach revealed that IED-associated variance in network activity in the epileptic DG is disproportionately correlated with the activity of abGC-dominated ensembles. This observation is consistent with the hypothesis that abGC-dominated ensembles drive IED-associated network activity. However, the experiments performed here cannot definitively exclude alternative hypotheses, such as the possibility that IED and ensemble heterogeneity are driven by variance in the origin of IEDs; testing such hypotheses could form the basis of future experiments.

6. The correlation matrices shown in Fig. S2E are problematic. The correlations are based on the fraction of abGCs in each ensemble and these fractions are sorted from low to high. The correlation will be inherently high by construction once you sort them and then obtain the correlation coefficient. Therefore, the main claim of this figure that "the statistics of this ensemble structure are highly conserved across animals" does not hold.

Thank you for this comment. Indeed, this is a limitation of the metric used, arising from the fact that there is no obvious way to establish correspondence of ensembles between animals other than by relative proportions of each population. We deliberately chose Pearson's correlation coefficient in order to attempt to measure excess correlation ascribable to shared structure beyond the sort itself, but the correlation that arises as a result of the matching procedure is indeed problematic. In light of these concerns, we have elected to remove this figure. However, this does not affect the other evidence for this claim, i.e., the consistency we see in other statistics of ensemble structure between animals, such as the fraction of abGC dominated ensembles, the "purity" of the purest abGC ensemble vs mGC ensemble, and the mean off diagonal correlations. Therefore, we believe that the claim still holds.

7. Terminology in Fig. 3C caption: 100-fold cross-validation on an 80-20 train-test split. Do the authors mean “100 times 5-fold cross-validation”? 100-fold would imply splitting data into 99% for training and 1% for test.

Thank you. We have fixed this caption.

Left: 100 times cross-validation on an 80-20 train-test split shows $85\pm 5\%$ classification accuracy on the test set under standard L2 regularization and $81\pm 5\%$ classification accuracy under L1 regularization

8. The mice are head-fixed and free to run on a treadmill belt. How is the animal's behavior related to the heterogeneity in the responses of abGCs and mGCs to individual IEDs? It is possible that certain degree of the observed heterogeneity arises from different behavioral states. For example, the responses of some abGCs or mGCs to IEDs may depend on if the animal is running or still, how fast the animals is running, etc. This is corroborated by that the most active ensemble can be decoded from the IED power spectrum (both could be influenced by animal's behavior) for small ensemble number. Unfortunately, this aspect is not analyzed at all.

Thank you for this comment. Behavior state modulation is indeed an important point to consider. For the purposes of this experiment, the animals were not trained to run. Therefore, there is relatively little behavior state heterogeneity. We do not find population-level differences in responsiveness to onset and offset of locomotion (**Fig. S1A**). Furthermore, the following plot (**Rebuttal Figure 2**) shows the raster of **Figure 4B** with running bouts highlighted in orange: running intervals account for less than 5% of the session.

Rebuttal Figure 2.

This is minor but the authors also implant an EMG electrode in the mice. How the EMG signal, which may reflect the animal's behavioral state, may explain the heterogeneity is not analyzed. It is also useful to check the relationship between the EMG signals and IEDs; is it possible that some of the IEDs and certain ensembles are correlated with EMG fluctuations? These control analyses would be helpful to exclude possibilities which could otherwise undermine the main conclusions.

Thank you. Unfortunately, while the EMG probe was implanted and is included in the Method for completeness, the signals recorded were not dynamic and were of low quality. As a result, these data were not used in the analysis.

9. In Table S1, please double check if the numbers are correct. The “Mean Total” seems wrong.

Thank you for pointing out this typo, the **Table S1** has been revised.

10. It is problematic that several of the conclusions are made on statistics from only one example animal (e.g. Fig. 2C, Fig. 3D) without analyzing the population statistics (all mice).

Thank you for the valuable feedback. We have now better integrated more population statistics into our results: we have replaced **Figure 2C** with the mean PSTH averaged across animals in the main text. In addition, we have added a further supplementary figure panel (**Figure S1E**) analyzing the population statistics of classification under the two penalties (**Rebuttal Figure 3**).

Rebuttal Figure 3 (Figures 2C and S1E).

11. Although the ratio between abGC-dominated and mGC-dominated ensembles are ~4:6 and the percentage of abGCs is only ~11% in the DG, this does not necessarily imply that the abGC-dominated ensembles contribute to the generation of IEDs more than mGC ensembles. It is possible that abGCs are simply more heterogeneous (therefore more distinguishable ensembles), or the majority of IEDs-related mGCs occupy a small number of ensembles; consequently, the total fraction of neurons involved in IEDs could be higher in mGCs than abGCs, disproving that abGC ensembles drive IEDs more prominently than mGC ensembles. Also, without further experiments, one could only say how GCs activity is correlated with but not how they could ‘drive’ IEDs.

Thank you for these important alternative hypotheses. It was not our intention to claim that more abGCs are involved in IEDs compared to mGCs *in absolute terms*. Indeed, statistically it is inevitable that more mGCs are involved in each IED, as mGCs far outnumber abGCs. However, we do make the claim that IED *heterogeneity* is reflected in abGC *ensemble heterogeneity*, as there are more abGC-dominated ensembles than one would expect by random sampling. We have addressed this important caveat and alternative hypotheses in the Discussion.

Our modeling approach revealed that, IED-associated variance in network activity in the epileptic DG is disproportionately correlated with the activity of abGC-dominated ensembles. This observation is consistent with the hypothesis that abGC-dominated ensembles drive IED-

associated network activity. However, the experiments performed here cannot definitively exclude alternative hypotheses, such as the possibility that IED and ensemble heterogeneity are driven by variance in the origin of IEDs; testing such hypotheses could form the basis of future experiments.

Furthermore, we have replaced this analysis with a more rigorous procedure for determining the number of abGC and mGC-dominated ensembles. This has been added to the Methods (page 30) as follows:

We identified “abGC dominated” and “mGC dominated” ensembles by sampling from a shuffle distribution as follows: The number and sizes of ensembles were fixed from the ensembles identified in real data. The cell type labels were then randomly permuted while preserving the overall number of abGCs and mGCs in the entire population. This procedure was repeated for 10,000 iterations to construct null distributions for the number of abGCs and mGCs expected in each ensemble based on its size. Ensembles which contained more abGCs or mGCs than the 95th percentile of its size-matched shuffle ensemble were termed abGC- or mGC-“dominated”, respectively.

12. How is the bootstrapped, binarized matrix z' constructed? Does it solely depend on other synchronous network activity excluding IEDs? This seems to be the case as in the caption of Fig. 4B. Then the ensemble structure is estimated from this matrix and the results are used for the analysis of IEDs events. Does it not depend heavily on the hypothesis that the same ensemble structure is preserved across other synchronous network activity and IEDs?

We do not see significant synchronous network activity outside of IEDs. However, we do verify that the ensemble structure is conserved across IEDs, as cells which are co-active in one IED tend to be co-active in other IEDs. In **Figure 4F**, we sort the cells according to ensemble, and find that cells within an ensemble are highly correlated to each other compared to cells outside that ensemble.

We have clarified the construction of the binarized matrix in the revised Methods section (page 29):

The responsiveness was binarized to an “activation” variable z' , using the following bootstrapping procedure: a bootstrap responsiveness distribution was calculated by drawing 10,000 random trigger frames with replacement from the $\Delta F/F$ trace of each cell, excluding IEDs and seizures. A cell was determined to be “activated” by an IED if its responsiveness to that IED exceeded the 95th percentile of the bootstrapped responsiveness distribution.

Reviewer #3 (Remarks to the Author):

The manuscript by Sparks et al., describes the relative contribution of the activity of adult born dentate granule cells and mature granule cells to interictal epileptiform discharges in the hippocampus. The authors use several novel techniques to classify which populations of cells are activated in response to any given IED, and compare ensemble activity across animals. The

authors find a set of latent ensembles of DGs involved in IED activity and these ensembles are disproportionately dominated by activity of abGCs. Comments are below:

1. My biggest concern with the manuscript is that it is unclear what these results mean for our understanding of epilepsy in general. For the composition of these ensembles to relate to epilepsy more broadly, it would be good to show that some result correlates with an epileptic phenotype. For example, the number of IEDs with pure abGC ensembles or the pureness of the ensembles in any given animal would relate to seizure frequency or duration on an animal-by-animal basis, or perhaps more tractably the ensemble composition could relate to the ability of the IED to generalize on an IED-by-IED basis. The assertion that the findings are an epilepsy-specific phenomenon and is not a function of how DG neurons behave in ensembles more broadly is also questionable. I do not see any effort made to distinguish DG ensemble activity in IEDs from activity in normal ongoing function or evoked stimulation. However, the technique used to characterize latent ensemble recruitment is interesting and novel.

Thank you for the insightful and helpful comments. The presence of IEDs is inherently pathological; therefore, we believe that any response of GCs to these pathological events is interesting for our understanding of epilepsy. By focusing on IEDs and GC activity, and building on earlier 2P imaging studies in vitro (Feldt Muldoon et al., 2013), we aimed to advance our understanding of how macroscopic events, as observed with LFP/EEG electrodes, relate to microscopic, cellular-level patterns of activity in chronically epileptic neuronal networks, which is considered to be a critically important outstanding problem of epilepsy research (reviewed in Farrell et al., 2019). As the reviewer points out, the technique used to characterize latent ensemble recruitment is novel, and it seems reasonable to expect that it will also be useful to investigate in future studies on the involvement of distinct DG ensembles in normal DG functions.

2. Have the authors used the classifier on snips of control LFP to see if it catches sharp waves? Can the authors use the movement information to inform whether they are capturing IEDs vs sharp waves?

Thank you for this comment. This is an important question but beyond the scope of the study, especially since the relationship between sharp waves in the CA1 and dentate GC ensemble activity appears to be a complex one, based on the few studies that have begun to address this interesting question (Penttonen et al., 1997; Headley et al., 2007; Pofahl et al., 2020, Dvorak et al., 2020). We did not aim to capture or image sharp wave events in control animals and recordings were restricted in TLE animals. Furthermore, while IEDs are readily detectable using a single-channel electrode (as in our study), reliable detection of sharp wave ripples would require multi-channel probes.

3. The authors should include a figure showing the LFP classifier feature space.

Thank you for the comment. We show the LFP classifier feature space schematically in **Figure 2A**. However, due to kernelization, the effective space in which classification occurs is infinite dimensional. The features that the kernel classifier receives are simply a magnitude spectrum, which we show in **Figure 2**.

4. How are the authors binning to classify a cell as responsive or non-responsive to an IED event, e.g. what is the time window for calcium changes in a cell in relation to an identified LFP IED that would classify that cell as “responsive”? How was this chosen? Over what period of time was activity averaged? Have the authors done a robustness analysis to validate their binning choice?

Thank you. We choose a time window of 3 seconds pre and post IED in order to capture fast and slow-timescale responses evoked in the slow calcium signal. We have performed a robustness analysis comparing this choice to 1 s of activity pre/post vs 5 s of activity pre/post, yielding a highly similar bootstrapped activation matrix z' compared to 3 s (**Rebuttal Figure 4**). Narrower windows consisting of only a few frames are likely to be less useful due to the slow sampling rate of the calcium, while wider windows are unlikely to be useful due to the decay time constant of calcium responses.

Rebuttal Figure 4.

The following text has been added to Methods (page 29):

A scalar responsiveness Δ of a cell to an event was calculated as the cell's mean activity in a fixed window pre-event subtracted from the cell's mean activity in the same window post-event. The analyses shown use a window size of 3 seconds, though the results do not change appreciably for window sizes between 1 second and 5 seconds (data not shown)

[...]

The responsiveness was binarized to an “activation” variable z' , using the following bootstrapping procedure: a bootstrap responsiveness distribution was calculated by drawing 10,000 random trigger frames with replacement from the $\Delta F/F$ trace of each cell, excluding IEDs and seizures, and then used to construct a 95% confidence interval. A cell was determined to be “activated” by an IED if its responsiveness to that IED exceeded the 95th percentile of the bootstrapped responsiveness

5. How are you determining the 0.1 cutoff for the IED mGC-ness score? What proportion of the IEDs have a sameness score between -0.1 and +0.1 (or is the histogram in 3B all the data from that animal)? Does the bi-modality hold for all animals?

Thank you. The intercept (0.1) is determined as part of the logistic regression fitting process. The histogram in **Figure 3B** represents all data from this animal. The bimodality holds for all animals. We have added a further pooled supplemental figure (**Fig S1E**) emphasizing the consistency of classification across animals.

6. The L1 regularization analysis seems tautological and difficult to cross validate. Can the authors show a scatterplot of responsive cells to each spike by cell type?

Thank you for the valuable comment. Each session may have several hundred spikes, and each cell's spike responsiveness profile is a vector of the same length. Due to random recruitment and the presence of mixed ensembles, there is high variability in cells' responsiveness to any individual spike. However, we can perform dimensionality reduction on the multi-hundred-dimensional cell responsiveness vectors using UMAP in order to visualize the responses of the populations to spikes (**Rebuttal Figure 5, Figure S1D**).

Rebuttal Figure 5 (Figure S1D).

We have also added the following section to Methods (page 29):

UMAP dimensionality reduction

We use supervised UMAP to perform dimensionality reduction in order to visualize the cell IED response profiles in high dimensional space. Dimensionality reduction was performed using the UMAP-learn package. The UMAP embedding was cross-validated in a similar manner to the logistic regression: UMAP was trained on a random 80% of the cells and the learned embedding was applied to a held-out on 20% of the cells.

7. Is there some feature of the IED that can be mapped on to whether the ensemble is a mCG-/ abGC-selective or a mixed ensemble?

Thank you for the insightful comment. In this work, we primarily focus on whether features of the low-dimensional global LFP signal can be decoded from the activities of individual cells and ensembles. The question of whether the (high-dimensional) population activity can be predicted from the one-dimensional global LFP signal, which is the “critical inverse problem” that we allude to, is a very scientifically interesting question which could form the basis of future work in this area, but is mostly outside the scope of this work. However, we show in **Figure 4 H** that indeed there is some information about the recruitment profile of an event in its LFP spectrum which is

decodable using Random Forests, a highly nonlinear “black box” decoding algorithm. All of the linear decoders we tried performed no better than chance.

We have expanded upon the Discussion in this regard (page 15):

In this work, we primarily investigated whether features of the low-dimensional global LFP signal can be decoded from the activities of individual cells and ensembles. The critical inverse problem of whether the (high-dimensional) population activity can be predicted from the one-dimensional global LFP signal is of great interest, though more difficult as it requires predicting a nonlinear statistic of high-dimensional data from a one-dimensional LFP signal. Our finding that, for small ensemble numbers, the most active ensemble in each IED could be decoded significantly above chance from the power spectrum of the IED alone suggests that specific GC ensembles do have distinct LFP signatures (see Valero et al., 2017 for another example). While there is no obvious way to equate ensembles across animals, whether these LFP signatures are conserved across animals poses an interesting open question for future work

8. Are data in figure 4A-F for one recording session in one animal? Do these results hold up across animals?

Thank you for the comment. We have clarified that the data in **Figure 4A-F** are examples from one animal. However, summary statistics of these results across animals are shown in **Figure S2**.

9. 4E, left is never described in the text

Thank you, we have moved this panel to supplement and we reference it in the main text (page 12)

10. What’s the cut off for an abGC- or mGC- “dominated” ensemble? Does this vary across animals? Does it vary as a function of the frequency of IEDs even within animal?

Thank you for the comment. We term an ensemble “dominated” by a certain cell type if that cell type is overrepresented in that ensemble compared to chance sampling. We have replaced this analysis with a more rigorous procedure for determining the number of abGC and mGC-dominated ensembles. This has been added to the Methods (page 30) as follows:

We identified “abGC dominated” and “mGC dominated” ensembles by sampling from a shuffle distribution as follows: The number and sizes of ensembles were fixed from the ensembles identified in real data. The cell type labels were then randomly permuted while preserving the overall number of abGCs and mGCs in the entire population. This procedure was repeated for 10,000 iterations to construct null distributions for the number of abGCs and mGCs expected in each ensemble based on its size. Ensembles which contained more abGCs or mGCs than the 95th percentile of its size-matched shuffle ensemble were termed abGC- or mGC-“dominated”, respectively.

11. Are data displayed in 4H from all IEDs in all animals? Most example traces show 4 or 6 ensembles and text states that K=6 is validated across animals, but the accuracy of decoding the IED is at chance for those ensemble sizes. Could the authors please discuss?

While the focus of this work is decoding in the “forward” direction (using cell activities to decode IED), we find that some level of decoding in the “inverse” direction is possible. Practically, the inverse problem is much more difficult to solve, as it entails predicting a nonlinear statistic of high dimensional data (namely, the cells active) from low-dimensional data (the global LFP signature). As the number of ensembles to be decoded in this way increases, the problem is compounded. We believe decoding of larger ensemble numbers would improve with more training data, though this is outside the scope of this current work.

We have added the following text to the Discussion (page 15):

In this work, we primarily investigated whether features of the low-dimensional global LFP signal can be decoded from the activities of individual cells and ensembles. The critical inverse problem of whether the (high-dimensional) population activity can be predicted from the one-dimensional global LFP signal is of great interest, though much more difficult as it requires predicting a nonlinear statistic of high-dimensional data from a one-dimensional LFP signal. Our finding that, for small ensemble numbers, the most active ensemble in each IED could be decoded significantly above chance from the power spectrum of the IED alone suggests that specific GC ensembles do have distinct LFP signatures (see Valero et al., 2017 for another example). The number of ensembles decodable from this low-dimensional signal does not necessarily reflect the number of ensembles in the network; rather, the complexity of decoding increases combinatorially with the number of ensembles to be decoded. We believe decoding of larger ensemble numbers would improve with more training data, though this is outside the scope of this current work. While there is no obvious way to equate ensembles across animals, whether these LFP signatures are conserved across animals poses an interesting open question for future work.

Minor

All animals are presumably recorded for multiple test sessions but overall, the reader has to work pretty hard to determine which figures are from one recording vs one animal vs pooled over the entire cohort. It is unclear why more pooled data are not shown, unless the results do not generalize?

Thank you for your helpful comments on the presentation of our results. The principal innovations presented in this work are conceptual, relating the activity of individual events to individual cells and ensembles. We feel that it is important to illustrate these results for the reader by example first before presenting extensive second-order statistics. However, we have supplemented **Figures 2-4** with pooled data as follows: We have replaced **Figure 2C** with a pooled figure from 5 animals. We have also supplemented **Figure 3** with panel **Figure S1E** showing that the classifiability of abGCs and mGCs by IED-responses generalizes across 5 animals. **Figure S2** shows summary statistics for the modeling procedure depicted in **Figure 4** across animals.

Red and blue are not labeled in Figure 3D

Thank you. Corrected

The organization on Figure 3 makes it hard to read

Thank you, we hope that with the reorganization of other figures, the presentation has now been improved.

4H is cited before 4G

Thank you. Corrected

4G orange and blue are not labeled

Thank you. Corrected

Figures 4A, B, D and S2A are so small, they're almost impossible to see
Enlarged, reorganized.

Thank you. Corrected

Was the location of the LFP recording electrode in the CA1 cell layer verified?

Thank you. Our electrode placement in the CA1 was based on stereotactic coordinates. Accurate placement of the single wire electrode was not restricted to the pyramidal cell layer, as detection of the macroscale IED events is not dependent on the electrode being in this layer (indeed, IEDs may be detected using surface EEG electrodes alone, Javidan, 2012). Therefore, we did not design the experiments in a way that required histological verification of precise electrode placement within CA1. Future studies will be able to use the analytical techniques presented in this paper to ask DG-related questions that do require the placement of LFP electrodes in the CA1 cell layer, e.g., regarding the precise relationship between replay of specific episodic memory sequences during ripples in the CA1 and the recruitment of DG GC ensembles in both control and chronically epileptic animals.

References

- DANIELSON, N. B., KAIFOSH, P., ZAREMBA, J. D., LOVETT-BARRON, M., TSAI, J., DENNY, C. A., BALOUGH, E. M., GOLDBERG, A. R., DREW, L. J., HEN, R., LOSONCZY, A. & KHEIRBEK, M. A. 2016. Distinct Contribution of Adult-Born Hippocampal Granule Cells to Context Encoding. *Neuron*, 90, 101-12.
- DVORAK, D., CHUNG, A., PARK, E. H., FENTON, A. A. Dentate spikes and external control of hippocampal function. *bioRxiv*. 2020 Jul 20.
- HEADLEY, D.B., KANTA, V., & PARE, D. 2017. Intra-and interregional cortical interactions related to sharp-wave ripples and dentate spikes. *Journal of neurophysiology*, 117(2), 556-65.
- IKRAR, T., GUO, N., HE, K., BESNARD, A., LEVINSON, S., HILL, A., LEE, H. K., HEN, R., XU, X. & SAHAY, A. 2013. Adult neurogenesis modifies excitability of the dentate gyrus. *Front Neural Circuits*, 7, 204.
- JAVIDAN, M. 2012. Electroencephalography in mesial temporal lobe epilepsy: a review. *Epilepsy Res Treat*, 2012, 637430.
- NAKASHIBA, T., CUSHMAN, J. D., PELKEY, K. A., RENAUDINEAU, S., BUHL, D. L., MCHUGH, T. J., RODRIGUEZ BARRERA, V., CHITTAJALLU, R., IWAMOTO, K. S., MCBAIN, C. J., FANSELOW, M. S. & TONEGAWA, S. 2012. Young dentate granule cells mediate pattern separation, whereas old granule cells facilitate pattern completion. *Cell*, 149, 188-201.
- PENTTONEN, M., KAMONDI, A., SIK A., ACSADY, L., BUZSAKI, G. 1997. Feed-forward and feed-back activation of the dentate gyrus in vivo during dentate spikes and sharp wave bursts. *Hippocampus*, 7(4), 437-50.
- POFAHL, M., NIKBAKHT, N., HAUBRICH, A.N., NGUYEN, T., MASALA, N., BRAGANZA, O., MACKE, J.H., EWELL, L.A., GOLCUK, K., BECK, H. 2020. Dentate gyrus population activity during immobility supports formation of precise memories. *bioRxiv*. 2020 Jan 1.
- SAHAY, A., SCOBIE, K. N., HILL, A. S., O'CARROLL, C. M., KHEIRBEK, M. A., BURGHARDT, N. S., FENTON, A. A., DRANOVSKY, A. & HEN, R. 2011. Increasing adult hippocampal neurogenesis is sufficient to improve pattern separation. *Nature*, 472, 466-70.

Reviewers' Comments:

Reviewer #1:

Remarks to the Author:

I have no further comments to the authors. All questions/criticism were sufficiently addressed.

Reviewer #2:

Remarks to the Author:

The authors have addressed most of my concerns. I have a few remaining comments:

1. The authors could use some better organization of the figures.
2. A couple of the responses to my previous comments are not reflected in the manuscript, e.g. Figure S2E, which is potentially misleading due to the way of sorting, is still mentioned in the main text.
3. Figures S1D&E, 2F are missing.
4. Line 289: reference to Figure 3D instead of Figure 3C; missing statistics when claiming 'significant' vs. 'not significantly'.

Reviewer #3:

Remarks to the Author:

The authors have addressed many of my comments. Changes to the figures and additions to the discussion have greatly improved the manuscript overall. However, my main concern still remains:

While I agree that the presence of IEDs is inherently pathological, SPW-Rs coexist with IEDs in the epileptic brain and IEDs can overlap with normal SPW-Rs in amplitude and duration (Buzsaki 2015). These may therefore be picked up by their IED algorithm, especially given that the IEDs in the kainite model can be heterogeneous (Chauviere et al., 2012) and that the method was not tested for "false-positives" with control LFPs. In light of this and the fact that there is no control DG network comparator in the study, in order to suggest that this is a pathological rather than a physiological feature of the DG network, it is important that some aspect of the observed DG cell networks relate to some feature of the epilepsy.

The authors should attempt to correlate any of the features of the functional organization of the DGC networks within each animal to the severity of the animal's pathology (e.g. seizure frequency, duration, etc.) or correlate the composition of the networks on an IED-by-IED basis to the "severity" of the IED (e.g. the duration the individual IED or its ability to generalize into a seizure.) The authors state in the methods that they have information on seizure frequency and duration and have information on every IED, so I believe both of these analyses should be possible with their current data set and would greatly strengthen statements about "pathological ensembles." Without these analyses, while I still remain enthusiastic about the latent ensemble recruitment method, it is extremely difficult for me to be enthusiastic about the biological results of the paper and feel it would be more appropriate for a methods journal.

Once again, we thank our Reviewers for considering the initial revisions that we made to the manuscript, and for further highly constructive comments that remained to be addressed. Our point-by-point responses to the remaining specific concerns are noted below with the Reviewers' comments retained in blue.

Reviewer #1 (Remarks to the Author):

I have no further comments to the authors. All questions/criticism were sufficiently addressed.

We thank the Reviewer for their efforts in assessing our manuscript and providing invaluable feedback which has improved the paper.

Reviewer #2 (Remarks to the Author):

The authors have addressed most of my concerns.

We thank the Reviewer for their efforts in assessing our manuscript and providing invaluable feedback which has improved the paper.

I have a few remaining comments:

1. The authors could use some better organization of the figures.

Thank you for this comment. To improve readability and flow, we have reorganized **Figures 1, 2, and 3**.

2. A couple of the responses to my previous comments are not reflected in the manuscript, e.g. Figure S2E, which is potentially misleading due to the way of sorting, is still mentioned in the main text.

3. Figures S1D&E, 2F are missing.

An intermediate version of the revised supplementary figures was mistakenly included when the submission was uploaded. We sincerely apologize for this error. The correct versions should reflect these changes.

4. Line 289: reference to Figure 3D instead of Figure 3C; missing statistics when claiming 'significant' vs. 'not significantly'.

Thank you for the comment. The figure reference has been corrected. Statistics have been added to verify the effectiveness of the classification procedure. The revised text reads as follows:

"A two-way analysis of variance yielded a main effect for IED class (pro-mGC vs anti-mGC, $F(3,54) = 14.11$, $p=6 \times 10^{-7}$), such that pro-mGC IEDs significantly positively modulated mGCs compared to anti-mGC IEDs. The main effect of population was non-significant ($F(3,54)=0.13$, $p=0.71$), which suggests that our observations cannot be explained by intrinsic differences in responsiveness between the two populations independent of the type of IED. However, the interaction effect was significant ($F(3,54) = 10.29$, $p=0.002$), indicating a crossing over effect, i.e., that pro-mGC IEDs significantly positively modulated mGCs over abGCs. This verifies that single cell responses to the

“most informative” IEDs identified by this procedure showed high within-population heterogeneity but striking between-population differences [...]”

Reviewer #3 (Remarks to the Author):

The authors have addressed many of my comments. Changes to the figures and additions to the discussion have greatly improved the manuscript overall. However, my main concern still remains:

While I agree that the presence of IEDs is inherently pathological, SPW-Rs coexist with IEDs in the epileptic brain and IEDs can overlap with normal SPW-Rs in amplitude and duration (Buzsaki 2015). These may therefore be picked up by their IED algorithm, especially given that the IEDs in the kainite model can be heterogeneous (Chauviere et al., 2012) and that the method was not tested for “false-positives” with control LFPs. In light of this and the fact that there is no control DG network comparator in the study, in order to suggest that this is a pathological rather than a physiological feature of the DG network, it is important that some aspect of the observed DG cell networks relate to some feature of the epilepsy.

Thank you for reinforcing the importance of adding this control to the manuscript, it not only strengthens the paper but increases the biological relevance. In response, we have now added an additional control experiment in which we specifically targeted hippocampal area CA1 with multisite silicone probe LFP monitoring and detection of sharp wave ripples (SPW-Rs) and concurrently imaged DG in the healthy brain in two animals. These population events serve as a control DG network comparator, and we find that the abGC and mGC population responses are synchronous during these events and comparatively desynchronized during IEDs in TLE. We have added these findings to the Results Section (page 13), and **Supplementary Figure 4**. The new text describing these results reads as follows:

“Sharp-wave ripples recruit synchronized mixed ensembles of abGCs and mGCs

To verify that these observations about ensemble structure are a feature of TLE rather than a physiological feature of the DG network in response to any electrophysiological event, we imaged DG while recording sharp-wave ripples (SPW-Rs) in stratum pyramidale of CA1 in non-epileptic mice (Figure S4A). We then used the LDA/LER model to examine the ensemble structure underlying SPW-Rs in the control DG. As in Figure 4, we compute a binarized “recruitment” matrix of cell activity that is significantly positively modulated within SPW-Rs compared to non-SPW-R epochs (see Methods). From the recruitment matrices, we observe qualitatively that the abGC and mGC populations appear to be activated more sparsely and more synchronously than in IEDs in TLE (Figure S4B). As before, we train the model on the binarized recruitment matrix (Figure S4C,D). We find that the non-epileptic DG does not segregate into abGC and mGC ensembles; instead, all inferred ensembles are mixed. Nonetheless, we do confirm that the learned ensembles correspond to real correlation structure in the underlying cells (Figure S4E). To quantify this desynchronization across animals, we calculated Kendall’s tau correlation between the median event responsiveness vectors of the two populations in each recording session, and find that abGCs are significantly less synchronous with mGCs in IEDs in TLE compared to in SPW-Rs in control ($p = 0.017$, Mann-Whitney

U-test on $N_{TLE}=17$ IED recording sessions from 5 mice, $N_{CTRL}=8$ SPW-R recording sessions from 2 mice; **Figure S4F**).

In addition, we have added the following text to the **Discussion** (page 16):

We also explored the implications of this model in the context of SPW-R related ensemble recruitment. We find that the abGC and mGC populations are more synchronous in SPW-Rs and organize into more sparsely activated mixed ensembles compared to IEDs. These observations suggest that physiological DG microcircuits use these two populations in conjunction to perform ripple-associated computations; in contrast, the emergence of desynchronized population-specific ensembles in TLE suggests that abGCs decoupling from mGCs and becoming coupled to each other may be one of the pathomechanisms of TLE. These observations are consistent with the hypothesis that abGC-dominated ensembles drive pathological IED-associated network activity. However, the experiments performed here cannot definitively exclude alternative hypotheses, such as the possibility that IED and ensemble heterogeneity are driven by variability in the origin of IEDs; testing such hypotheses could form the basis of future experiments.

Figure S4: Ensemble recruitment by sharp wave ripples in control animals. **A.** Schematic of simultaneous 2p imaging and SPW-R recording. **B.** Binarized activation matrix with respect to SPW-Rs, using the same binarization procedure as in Figure 4B. **C.** Ensemble activity matrix shows sparse activation of ensembles. **D.** Learned ensembles in the network as in Figure 4D. The non-epileptic

DG does not segregate into abGC and mGC ensembles; instead, all inferred ensembles are mixed, suggesting that physiological microcircuits use the two populations to perform ripple-associated computations in a synchronized manner, which is disrupted in IEDs in TLE. E. Cell-cell correlation matrix, sorted by ensemble ID. Cells tend to be correlated with other cells in the same ensemble, while having low correlation with cells in different ensembles. F. abGC event responses are significantly desynchronized from mGCs event responses in TLE. Synchrony is defined as the Kendall tau correlation between event-triggered activations in each population (Mann-Whitney U-test on $N_{TLE}=17$ IED recording sessions from 5 mice, $N_{CTRL}=8$ SPW-R recording sessions from 2 mice, $p = 0.017$).

In addition, we have added related text to the **Methods Section** (pp. 27, 29):

“Detection of sharp wave ripple events

LFP signals were calculated from the wide-band 25 kHz signal by down-sampling to 1250 Hz. From the 4 channels on the probe, the wire within the pyramidal layer was identified as showing the greatest ripple-band Gabor wavelet power (100 to 225 Hz). After the removal of noisy LFP epochs, SWR events were detected using a custom Matlab supervised algorithm based on a template hand-labeled ground-truth data set of SWR events from 4 mice (separate from those recorded in this study) using *k*-nearest-neighbor embedding based on wavelet-derived SWR features. Candidate events identified using the supervised template matching procedure, were considered SWR-events if their within-event ripple-band wavelet power was at least 12 median absolute deviations (m.a.d.) above the median for the session. SWR-event detection was visually verified. Only SWR-events occurring during periods of immobility lasting at least 3 seconds were included in the analysis. For calculations requiring point time estimates, the within-SWR ripple-power peak was used.

Synchrony

The “synchrony” of abGCs and mGCs is defined as the Kendall tau correlation between the median event-activation vector of each population. Kendall’s tau was chosen as a “soft” correlation able to tolerate small displacements in time, meaning desynchronization in time is penalized on a continuous basis rather than as an all-or-nothing correlation as with e.g. Pearson’s *r*.

The authors should attempt to correlate any of the features of the functional organization of the DGC networks within each animal to the severity of the animal’s pathology (e.g. seizure frequency, duration, etc.) or correlate the composition of the networks on an IED-by-IED basis to the “severity” of the IED (e.g. the duration the individual IED or its ability to generalize into a seizure.) The authors state in the methods that they have information on seizure frequency and duration and have information on every IED, so I believe both of these analyses should be possible with their current data set and would greatly strengthen statements about “pathological ensembles.” Without these analyses, while I still remain enthusiastic about the latent ensemble recruitment method, it is extremely difficult for me to be enthusiastic about the biological results of the paper and feel it would be more appropriate for a methods journal.

Thank you for this insightful comment. We confirm that our observations about ensemble structure represent a pathological rather than a physiological feature of the DG network by comparing the ensemble structure within IEDs to the ensembles recruited by SPW-Rs, and find significantly reduced event-associated synchrony between the abGC and mGC

populations in TLE, as above. Furthermore, we have now also attempted to correlate this abGC-mGC desynchronization with a chronic disease phenotype, namely, seizure AUC (an electrographic measure of seizure severity) (**Rebuttal Figure 1**).

Regression lines fit to either IED synchrony or “event synchrony” (also including non-epileptic mice) vs seizure area-under-curve (AUC) in home-cage vEEG recordings suggest an inverse relationship between abGC-mGC event synchrony and severity of the epileptic phenotype. The inverse relationship we find between seizure AUC and abGC-mGC synchrony is consistent with our finding that the two populations are significantly desynchronized in IEDs in TLE compared to in SPW-Rs in the nonepileptic animal. Two drawbacks of the requested analysis should be noted: First, the correlation of an imaging-associated metric and a chronic vEEG metric can only be performed on a per-animal basis. Second, the goal of 24 hour vEEG monitoring described was to verify that animals had entered the chronic phase of TLE, which occurred approximately 10 days prior to the imaging sessions described in the study, and was halted once this occurred. Consequently, the true association between gold-standard chronic measures of disease severity and the correlation structure of the network may be higher, but we do not have more proximal data available in this dataset. Nevertheless, if the Reviewer considers that it is relevant and informative, we would be happy to add this panel as **Figure S4G**.

Rebuttal Figure 1

Rebuttal Figure 1. Seizure AUC, area-under-curve, was taken as the integral of an indicator variable representing seizure occurrence over the most proximal 24 hour period to the imaging session (10

days prior to imaging), which corresponds to the seizure frequency weighted by seizure duration in home cage in the 24 hours at that time point.

Relationship between abGC-mGC synchrony and seizure AUC in the most recent 24 hours of video EEG recorded prior to imaging, 1 point = 1 animal (orange points, $\Delta t = 10$ days). Synchrony for control animals (green) also plotted for comparison (AUC = 0). Regression lines fit to either IED synchrony only (orange dashed line, $y = -0.07 x + 0.66$, $r^2=0.32$, $p=0.32$) or "event" (IED or SPW-R) synchrony (blue dashed line, also including non-epileptic mice, $y=-0.07 x + 0.65$, $r^2=0.43$, $p=0.11$) suggest an inverse relationship between abGC-mGC event synchrony and severity of the epileptic phenotype.

Reviewers' Comments:

Reviewer #3:

Remarks to the Author:

The authors, Sparks et al., have addressed most of my comments. The inclusion of controls is appreciated; this very much strengthens the idea that these DG ensembles are a feature of a pathological network. If possible, please include this information in one of the highlights (e.g. "AbGC-dominated ensembles are disproportionately represented among the inferred ensembles during IED but not SW-R events")

Minor changes:

Please show the distribution of the synchrony data in S4D, rather than just a bar chart.

I believe there is utility to figure S4G and it is worth including in the final manuscript. It appears that, while synchrony is somewhat related to the seizure severity on a per-animal basis, it certainly does not explain a lot of the variability. This could very well be due to the caveats the authors listed in the rebuttal letter.

Reviewer #3

The authors, Sparks et al., have addressed most of my comments. The inclusion of controls is appreciated; this very much strengthens the idea that these DG ensembles are a feature of a pathological network. If possible, please include this information in one of the highlights (e.g. “AbGC-dominated ensembles are disproportionately represented among the inferred ensembles during IED but not SW-R events”)

We thank the Reviewer for their diligence in reviewing our manuscript. We believe the manuscript has been significantly improved as a result. We are advised by the Editors that it is not possible to include Highlights, but we have added similar language to the Abstract.

Minor changes:

Please show the distribution of the synchrony data in S4D, rather than just a bar chart.

We now show the distribution in a box and whisker plot.

I believe there is utility to figure S4G and it is worth including in the final manuscript. It appears that, while synchrony is somewhat related to the seizure severity on a per-animal basis, it certainly does not explain a lot of the variability. This could very well be due to the caveats the authors listed in the rebuttal letter.

Thank you for your recommendation. We have included Figure S4G.